# Long history paddy rice mapping across Northeast China with deep learning and annual result enhancement method

Zihui Zhang[1,2], Lang Xia[1,2], Fen Zhao[1,2], Yue Gu[3], Jing Yang[1,2], Yan Zha[1,2], Shangrong Wu[1,2], Peng Yang[1,2]

[1]State Key Laboratory of Efficient Utilization of Arid and Semi-arid Arable Land in Northern China, the Institute of Agricultural Resources and Regional Planning, Chinese Academy of Agricultural Sciences, Beijing, 100081, China
[2]Institute of Agricultural Resources and Regional Planning, Chinese Academy of Agricultural Sciences/Key Laboratory of Agricultural Remote Sensing, Ministry of Agriculture and Rural Affairs, Beijing, 100081, China
[3]Department of Electrical and Computer Engineering, Rutgers University, New Brunswick, OK 08901, USA

*Correspondence to*: Lang Xia (xialang@caas.cn); Fen Zhao (zhaofen@caas.cn); Peng Yang (yangpeng@caas.cn)

**Abstract** Northeast China, a significant production base for paddy rice, has received lots of attention in crop mapping. However, understanding the spatiotemporal dynamics of paddy rice expansion in this region remains limited, making it difficult to track the changes in paddy rice planting over time. For the first time, this study utilized multi-sensor Landsat data and a deep learning model, the full resolution network (FR-Net), to explore the annual mapping of paddy rice for Northeast China from 1985 to 2023 (available at https://doi.org/10.6084/m9.figshare.27604839.v1, Zhang et al., 2024). First, a cross-sensor paddy training dataset comprising 155 Landsat images was created to map the paddy rice. Then, we developed the annual result enhancement (ARE) method, which considers the differences in category probability of FR-Net at different stages to diminish the impact of the limited training sample in large-scale and across-sensor paddy rice mapping. ARE integrates differences in category probability and confidence levels of the FR-Net across phenological stages, effectively reducing classification uncertainty. This approach could mitigate the impact of limited training sample on large-scale and across-sensor paddy rice mapping. The accuracy of the paddy rice dataset was evaluated using 107954 ground truth samples. In comparison to traditional rice mapping methods, the results obtained using the ARE method showed a 5% increase in the F1 score. The overall mapping result obtained from the FR-Net model and ARE methods achieved high average values of user accuracy (UA) of paddy, producer accuracy (PA) of paddy, overall accuracy (OA), F1 score, and Matthews correlation coefficient (MCC) of 0.93, 0.91, 0.91, 0.92, and 0.82, respectively. The study revealed that the area used for paddy rice cultivation in Northeast China increased from $1.11 \times 10^4$ km$^2$ to $6.45 \times 10^4$ km$^2$ between 1985 to 2023. Between 1985 and 2023, there was an overall expansion of $5.34 \times 10^4$ km$^2$ in the paddy rice cultivation area, with the highest growth ($4.33 \times 10^4$ km$^2$) occurring in Heilongjiang province. This study shows that long-history crop mapping could be achieved with deep learning, and the result of paddy rice will be beneficial for making timely adjustments to cultivation patterns and ensuring food security.

# 1 Introduction

Paddy rice is an essential cereal crop globally and serves as the staple food for over half of the world's population (FAO, 2023). Therefore, it is crucial to gather accurate information on the temporal-spatial evolution characteristics and the long history of rice-growing districts. This knowledge can help us understand the intrinsic reasons affecting the temporal-spatial evolution in rice cultivation (You et al., 2021) and facilitate more efficient cultivation practices by adjusting the patterns of rice cultivation. Satellite remote sensing technology has been extensively utilized for mapping paddy rice on a regional or global scale, offering advantages over traditional ground-based monitoring methods (Yang et al., 2024). Based on the difference in spatial resolution, satellite data can be classified as low, medium, and high spatial resolution data. Low-resolution data, such as AVHRR and MODIS, provide long-history monitoring data but have limited spatial resolution and cannot recognize detailed spatial features (Xiao et al., 2005; Luo et al., 2020). High spatial resolution data, including GaoFen, QuickBird, IKONOS, etc., can recognize detailed spatial features on a meter scale. However, due to limited coverage availability and revisiting recycling, they are unsuitable for large-scale and long-history paddy rice monitoring. Medium-resolution data, such as Landsat, and Sentinel-2, can provide reasonable revisit recycling and sufficient spatial resolution in agriculture applications, making them widely used in crop mapping (Graesser and Ramankutty, 2017; Deines et al., 2019; Griffiths et al., 2020; Sun et al., 2021).

Commonly used paddy rice mapping methods can be categorized into phenology-based methods and data-driven algorithms. The phenology-based method involves using a spectral curve or index to illustrate the phenological differences between paddy rice and other crops (Ashourloo et al., 2022). While this method is simple and effective, cloud cover can cause gaps in satellite data during the growth period, increasing the difficulty to track phenological changes accurately. This limitation can reduce the accuracy of long-term paddy rice mapping (Carrasco et al., 2022; Dong et al., 2015). Data-driven algorithms include traditional machine learning and deep learning methods. Traditional machine-learning algorithms usually rely on manual feature extraction, which may be less effective in feature abstraction (Goldberg et al., 2021; Khojastehnazhand and Roostaei, 2022). Particularly in complex cropping systems, traditional machine-learning algorithms may fail to differentiate between paddy rice and other crops (Zhong et al., 2019; Kamir et al., 2020; Gao et al., 2023).

The demand for accurate and efficient crop mapping is increasing as computer technology advances rapidly. Deep learning models such as RNN (Recurrent Neural Network) (Thorp and Drajat, 2021) and semantic segmentation networks (Yang et al., 2022) are increasingly being used for crop identification (Akkem et al., 2023). RNN can remember and recurse, allowing it to capture the context in sequential data, thus abstracting the feature information of temporal remote sensing images and performing crop mapping (Kong et al., 2019). However, RNN requires continuous clear-sky satellite images as input, which is difficult to obtain due to cloud contamination. Under a cloud scene, the sequence feature learned by the model is different from the test dataset, leading to decreased performance in crop mapping (Chen et al., 2021; Akkem et al., 2023). Semantic segmentation is a computer vision task that aims to assign category labels to individual pixels in an image, achieving pixel-level classification (Lu et al., 2023; Sun et al., 2023). Compared with the RNN, the semantic segmentation model can automatically learn spectral and spatial features from one or more satellite images for end-to-end classification, without relying

on the sequence feature behind the time-series data. Therefore, the input data of the semantic segmentation model is more flexible (Gao et al., 2023; Lu et al., 2023), making it more suitable for large-scale crop mapping with a long history (Yang et al., 2022).

The semantic segmentation model offers advantages for large-scale mapping. However, determining the final mapping result for a specific year from multiple intermediate maps remains a challenge for large-scale paddy rice mapping (Feng et al., 2023). Paddy rice exhibits distinct spectral and spatial characteristics at different growth stages, making it difficult to obtain enough training samples to cover the full range of phenology on a large scale. In this situation, the trained model may fail to learn the features presented in the test dataset on a large scale, and the results generated by the semantic segmentation model under different phenology in one year may present different accuracies ( Zhang et al., 2014; Xia et al., 2022). In practice, the commonly used method to obtain yearly final results is to average or overlay the different paddy rice results within the year (Feng et al., 2023). Unfortunately, this approach simply merges paddy rice results in different phenology and does not consider the most accurate paddy rice results in different phenology. As a result, the final annual paddy rice result would inherit the errors from the paddy rice results in different phenology.

The significant increase in rice cultivation in Northeast China has caught the attention of researchers who are interested in understanding the spatial and temporal distribution of rice in the region. The area used for growing paddy rice has expanded by 144 % from 2000 to 2017 (Xin et al., 2020), and the northern boundary of paddy rice cultivation has shifted about 25 km northward from 1984 to 2013 (Liang et al., 2021). Despite various efforts to map paddy rice cultivation (Table 1), there is still a lack of detailed spatial data for the entire Northeast China. This data gap is hindering the accurate assessment of crop methane emissions and the development of sustainable agricultural policies.

**Table 1 Relevant mapping datasets of paddy rice in China**

| Coverage of Northeast China | Time range/Span | Data source | Spatial resolution | Reference |
| --- | --- | --- | --- | --- |
| Partly | 1990–2020/5years | Landsat | 30 m | (Zhang et al., 2023) |
| Partly | 2015/yearly | Sentinel-1 | 10 m | (Onojeghuo et al., 2018) |
| Partly | 2020/yearly | MODIS | 500 m | (Shao et al., 2023) |
| Partly | 2020/yearly | GF-6 WFV | 16 m | (Guo and Ren, 2023) |
| Partly | 2013–2021/yearly | Landsat | 30 m | (Xuan et al., 2023) |
| Fully | 2000–2017/yearly | MODIS | 500 m | (Xin et al., 2020) |
| Fully | 2017–2019/yearly | Sentinel-2 | 10 m | (Ni et al., 2021) |
| Fully | 2017–2022/yearly | Landsat | 30 m | (Shen et al., 2023) |

In this study, we focused on paddy rice in Northeast China and used long-term Landsat data from different sensors to track the annual changes in paddy rice fields for the first time. The contribution of this study is (1) constructing a cross-sensor training dataset for paddy rice using Landsat 5 TM and Landsat 8 OLI sensors, (2) proposing an annual results enhancement (ARE) method based on category probability to improve annual mapping result of paddy rice under different phenology within a year, and (3) generating yearly paddy rice maps with 30m spatial resolution in Northeast China from 1985 to 2023 for the first time.

These consistent paddy rice maps can be utilized for monitoring paddy rice dynamics and assessing the effects of land-use policies.

## 2 Materials and Methods

### 2.1 Study area

The study area (38°43′8.4″-53°33′50.4″N, 111°8′38.4″-135°5′45.6″E) shown in Fig.1(a) is located in Northeast China, which includes Heilongjiang province, Jilin province, Liaoning province, and northeastern Inner Mongolia. It covers approximately $1.26×10^6$ km$^2$, accounting for 13.13% of China's total area (Zhang et al., 2014; Xin et al., 2020). Northeast China experiences a frigid zone with a continental monsoon climate and an average elevation of 450 m. The average annual temperature is about 4.37°C, and the average annual precipitation is about 800 mm. One crop per year is cultivated in this region under these hydrothermal conditions (Xin et al., 2020). The primary crops are rice (Zheng et al., 2023), maize (Shi et al., 2022), and soybeans (Huang et al., 2022), each with distinct growth periods. Paddy rice production in Northeast China consistently accounts for over 20% of the total paddy rice production in China. The cultivation patterns of paddy rice have undergone significant transformations, becoming a prominent driver of land use change in the study area (Griffiths et al., 2019; Jin and Zhong, 2022).

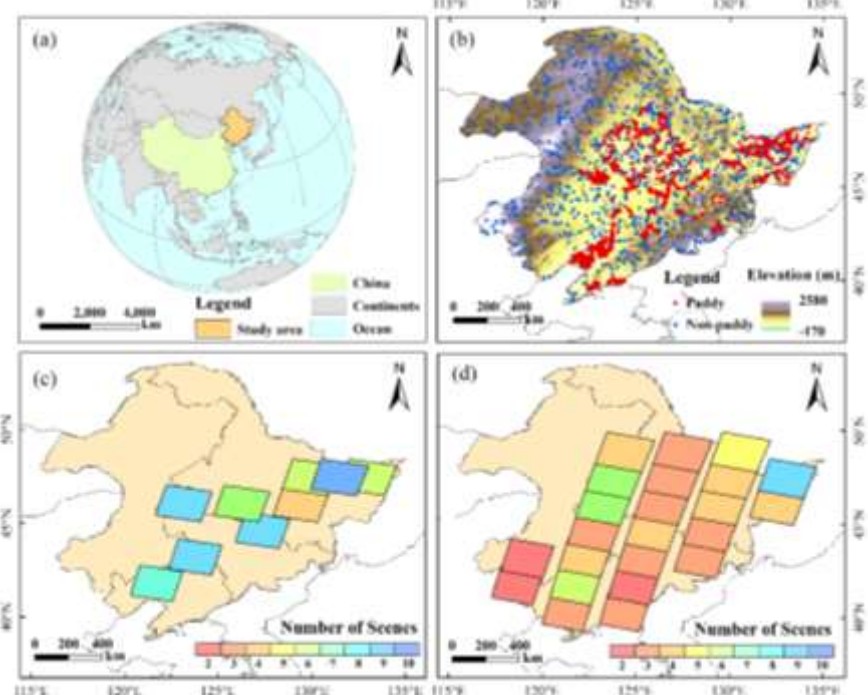

**Figure 1: (a) Location of the study area. (b) Distribution of ground truth data in Northeast China. (c), (d) are the number of training datasets for Landsat 5 and Landsat 8/9, respectively.**

## 2.2 Methods

### 2.2.1 Workflow of the study

The workflow for paddy rice mapping in Northeast China is illustrated in Figure 2. Firstly, Landsat data, paddy and non-paddy samples derived from Google Earth data and field survey, and the paddy cultivation area from agricultural statistics data between 1985 and 2023 were compiled to validate the accuracy of the annual paddy rice mapping results in this study. Secondly, a cross-sensor dataset containing 115 scenes paddy rice maps was generated using the XGBoost classifier combined with manual visual correction. This cross-sensor dataset served as training and testing datasets for developing the FR-Net model. Thirdly, based on the cross-sensor dataset, we employed the FR-Net and ARE methods to account for category probability differences across different phenological periods within a year to reconstruct annual paddy rice maps for Northeast China from 1985 to 2023, and the paddy rice maps were systematically validated with validation data. Finally, the paddy rice mapping results in this study were compared with representative products, and we analyzed the spatial and temporal dynamic characteristics of paddy rice in Northeast China.

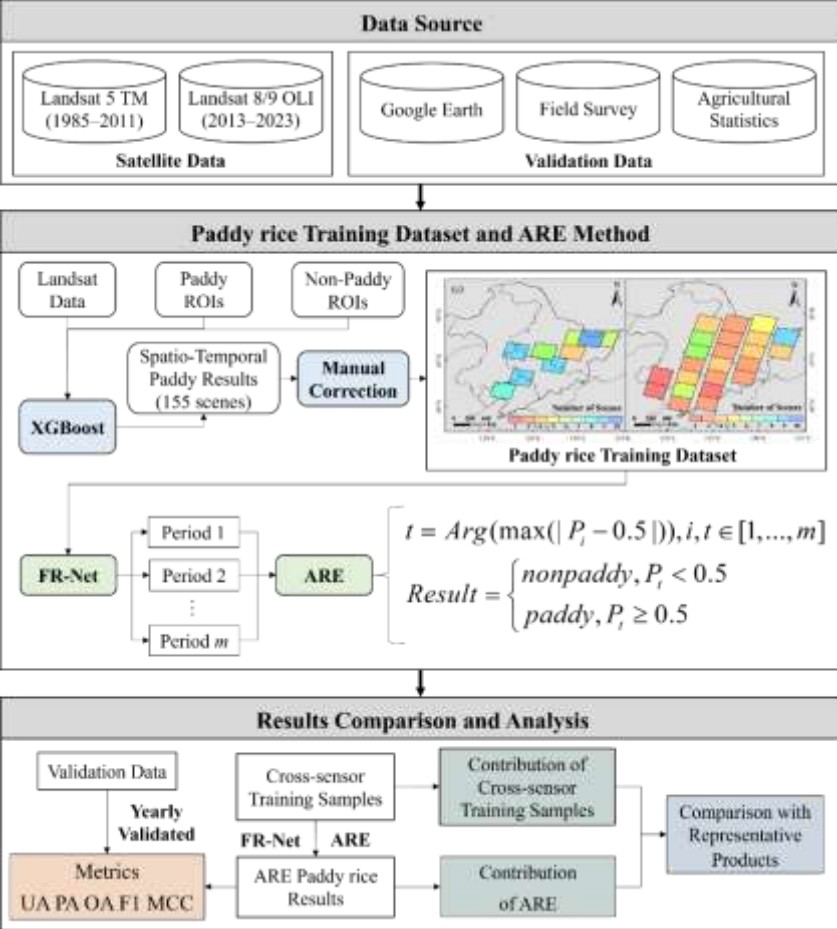

**Figure 2: Workflow of the study.**

### 2.2.2 FR-Net

A Full-Resolution network (FR-Net) was proposed to tackle the problem of low accuracy in edge segmentation caused by the loss of spatial information in deep semantic segmentation networks. The multi-resolution feature fusion unit (MRFU) serves as the core component of FR-Net, specifically designed to achieve high-resolution semantic segmentation while maintaining precise output quality. The MRFU regulates feature propagation through controlled information flow, integrates multi-scale feature representations via resolution-specific streams, and preserves spatial fidelity through hierarchical resolution retention.

Its architecture comprises two distinct pathways: the horizontal stream, which preserves native resolution through identity mapping operations, and the vertical stream, which doubles channel capacity while halving spatial resolution. Additionally, the MRFU incorporates 3×3 convolutional layers with a stride of 2, a batch normalization (BN) layer, and a rectified linear unit (ReLU) activation layer. These components work together to control and fuse feature streams with different resolutions. For the specific structure and implementation of FR-Net and MRFU, please refer to the published article by Xia et al. (2022).

FR-Net has a simple structure and requires minimal computational resources, making it suitable for extracting characteristic information from Landsat data and mitigating the issue of gradient disappearance. However, despite its straightforward design, the cascading operation of multi-resolution feature fusion may result in computational delays, which could hinder its ability to meet the near real-time requirements for agricultural monitoring. In this study, all FR-Net models utilized the same set of parameters for subsequent analysis.

### 2.2.3 Annual results enhancement method

During the growth period of paddy rice, multiple Landsat images of the same area can be obtained, allowing for the identification of each image and resulting in multiple mapping results. The commonly used method (Graesser and Ramankutty, 2017) to obtain the final mapping results from multiple mapping results within a year in the current large-scale semantic segmentation model for single paddy rice calculated following Eq.(1):

$$Result_{pre} = \begin{cases} nonpaddy. else \\ paddy, \forall i \in [1, \dots m]: Result_i = paddy \end{cases}, \tag{1}$$

where $Result_{pre}$ is the final annual mapping results, $m$ means the number of images during the paddy rice phenological period within the year, $i$ refer to order number of images, $Result_i$ is the result from the deep learning model, which includes two categories: paddy and non-paddy. Eq.(1) states that the output results of the deep learning model indicate that if there is a mapping result for any paddy period, the result for that specific year is classified as paddy; otherwise, it is classified as non-

paddy.

In large-scale paddy rice mapping, there are noticeable differences in the spectral and texture characteristics of the rice at different growth stages throughout the year (Yin et al., 2020; Pan et al., 2021). Furthermore, the high cost of generating high-quality training samples presents a significant challenge in ensuring comprehensive coverage of diverse phenological periods (Yeom et al., 2021). Therefore, errors may vary in paddy rice results for different phenological periods within the year. Simply

employing equation (1) would overlook this, perpetuate errors in different phenological periods, and reduce the accuracy of the final paddy rice map. Therefore, based on the distinct differences in spectral and texture characteristics of paddy rice across growth stages, we developed an annual result enhancement (ARE) method to address this limitation. ARE integrates differences in category probability and confidence levels of the FR-Net across phenological stages, effectively reducing classification uncertainty. This approach mitigates the impact of limited training sample on large-scale and across-sensor paddy

rice mapping. This method differs from previous studies, as demonstrated in Eq.(2):

$$t = Arg\big(max(|P_i - 0.5|)\big), i, t \in [1, \ldots, m]$$

$$Result = \begin{cases} nonpaddy, P_t < 0.5 \\ paddy, P_t \geq 0.5 \end{cases}, \tag{2}$$

where $i$ refers to the order number of images within the growth period, $P_i$ is the category probability output by FR-Net for images $i$, $max()$ means to obtain the maximum, $Arg()$ means to get the order number of images, $t$ represents the image

corresponding to the highest $P_i$ among $m$ images, $P_t$ is the category probability output by FR-Net for images $t$, $Result$ is the final paddy rice map within the year.

The ARE method considers the difference in category probability when mapping results show different phenology. It identifies the most accurate annual paddy rice mapping result by choosing the highest category probability among different mapping results. This approach can reduce the impact of feature differences between training and test data sets caused by limited training

samples on large-scale mapping. It also enhances the annual paddy rice mapping accuracy of the semantic segmentation model.

## 2.3 Dataset and processing

### 2.3.1 Acquisition and processing of Landsat images

A total of 13809 Landsat Collection 2 Level-2 surface reflectance products (30 m spatial resolution) covering Northeast China were acquired from the United States Geological Survey (USGS). These images were collected during the paddy rice growth

period (May to September) from 1985 to 2023, were utilized for paddy rice mapping, as illustrated in Fig.3. These images included Landsat 5 Thematic Mapper (TM) data from 1985 to 2011, as well as Landsat 8/9 Operational Land Imager (OLI) data from 2013 to 2023. Landsat 7 was adversely affected by stripes, resulting in poor data quality, hence the lack of 2012 imagery acquisition, processing, and analysis. In addition, we selected the Blue, Green, Red, Near Infrared (NIR), Shortwave Infrared (SWIR) 1, and Shortwave Infrared (SWIR) 2 bands of Landsat 8/9 OLI and Landsat 5 TM images to map the paddy

rice.

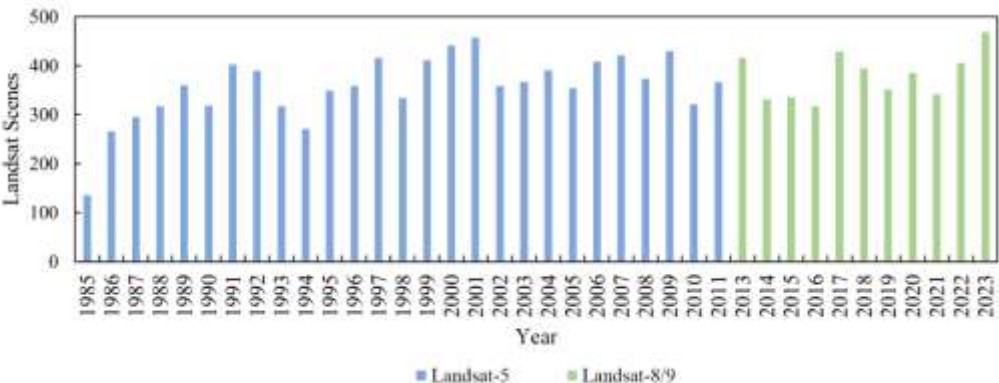

**Figure 3: Number of Landsat images from 1985 to 2023 used in this study.**

The training and validating data were obtained from the paddy mapping results based on eXtreme Gradient Boosting (XGBoost) maps after manual visual correction, as described below. First, we identified paddy and non-paddy regions of interest (ROIs) in Landsat images, and used these ROIs to train the XGBoost model. The XGBoost model was then utilized to initially obtain the temporal spatial distribution of paddy rice in the selected images. After that, manual correction was conducted to correct the temporal results of XGBoost. This process resulted in a high-precision paddy rice mapping dataset that covers the years from 1985 to 2023 with a total of 155 scenes (Fig.1(c), (d)). This dataset is divided into training and validation sets for the FR-Net model in a 3:1 ratio (Supplementary Table 1).

We have generated the training and validation labels for the FR-Net as below (available at https://doi.org/10.6084/m9.figshare.28283606, Zhang, 2025). Both the Landsat images and their corresponding masks were rotated by 5°. Subsequently, all the Landsat images and masks were cropped into several small images with dimensions of 256 × 256. These small images covered the Landsat images completely without any overlap, and any cropped image without paddy rice pixels was removed. The training and validation sets for Landsat 5 images were 29906 and 9968, respectively, the training and validation sets for Landsat 8/9 images were 50956 and 16985, respectively.

### 2.3.2 Ground truth dataset

The ground truth dataset was collected from field surveys and Google Earth imagery across Northeast China. Due to the unavailability of very high resolution (VHR) data before 2002, the Google Earth VHR imagery was limited to the period from 2002 to 2023, while field survey data were collected from 2011 to 2023. Paddy data from the field survey were identified using a digital camera and a Global Positioning System (GPS) receiver (Garmin GPSMAP 78s) with an accuracy of ±3 m (Xia et al., 2022). To ensure comprehensive representation of the validation dataset, the study area was stratified into major paddy cultivation areas and non-paddy regions. Ground truth data were randomly sampled from each stratum, maintaining an approximate 3:2 ratio between paddy and non-paddy areas. This resulted in a total of 68856 paddy samples and 39098 non-paddy samples, as illustrated in Fig.1(b). Specifically, the dataset includes 21254 paddy samples and 13160 non-paddy samples from field survey, complemented by 47602 paddy samples and 25938 non-paddy samples from Google Earth VHR imagery.

To mitigate the effects of mixed pixels in the validation results, detailed field observation data were conducted within a 30m × 30m square around each sample point, recording the proportions of the coverage for different land cover types. For samples containing multiple land cover types, the final classification was assigned based on the predominant type. Given the spatially extensive agricultural fields typical of Northeast China, a dominance threshold of 50% was applied to determine the primary land cover type. The field survey period was synchronized with the paddy rice growing season (May to September) to correspond with the satellite image acquisition period. This comprehensive dataset was used to evaluate the accuracy and reliability of the paddy rice mapping results.

## 2.4 Agricultural statistical data during 1985-2022

We have utilized agricultural statistical data obtained from district, municipal, and provincial statistical bureaus dating from 1985 to 2022. The statistical data we used includes paddy planted areas, which helped us confirm the accuracy and reliability of the paddy rice maps we provided for Northeast China. This confirmation was based on the total area under cultivation per year and temporal variations. As agricultural statistical data for 2023 are not yet available, we validated the paddy rice maps from 1985 to 2022. The accuracy of the paddy rice map in 2023 was verified using the ground truth data obtained in 2023.

## 2.5 Model training and accuracy assessment

To mitigate class imbalance during model training, the Dice loss function was employed in this study. This metric, derived from the Dice similarity coefficient (DSC; Eq.3), demonstrates inherent robustness against skewed class distributions by equivalently weighting false positive and false negative errors during optimization, thereby addressing prevalent challenges in imbalanced semantic segmentation tasks.

$$DSC = \frac{2 \times |A \cap B|}{|A| + |B|},$$ (3)

where $|A \cap B|$ quantifies the intersection cardinality between the predicted paddy rice pixels (A) and ground truth paddy rice pixels (B); $|A|$ and $|B|$ represent the quantity of paddy rice pixels in A and B, respectively.

The model was implemented on a workstation equipped with a single NVIDIA GeForce RTX 3090 GPU, Intel i5-13400K CPU, and 1 TB SSD. We installed Keras 2.5, Tensorflow 2.6, CUDA 11.3, and cuDNN 8.2 on the workstation for model training. The Adam with a constant learning rate of 0.001 was selected as the optimizer, and the batch size was 8. The model was trained five times, and the one with the best validation accuracy was used for mapping the paddy rice.

To reduce the influence of mixed pixels, we represented the confusion matrix based on the area ratio. In this study, we used the user accuracy (UA), producer accuracy (PA), overall accuracy (OA), F1 score, and Matthews correlation coefficient (MCC) to evaluate the accuracy of the long history paddy rice maps, as shown in Eq.(4)–(8):

$$UA = \frac{TP}{TP + FP},$$ (4)

$$PA = \frac{TP}{TP + FN},$$ (5)

$$OA = TP + TN, \tag{6}$$

$$F1\ score = 2 \times \frac{UA*PA}{UA+PA}, \tag{7}$$

$$MCC = \frac{TP \times TN - FP \times FN}{\sqrt{(TP+FP)(TP+FN)(TN+FP)(TN+FN)}}, \tag{8}$$

where TP is the proportion of true positive, TN is the proportion of true negative, FP is the proportion of false positive, and FN is the proportion of false negative. UA, PA, OA, F1 score, and MCC are commonly used metrics for evaluating classification accuracy (Foody, 2020; Olofsson et al., 2014; Xu et al., 2023), and F1 score is a harmonic mean of precision and recall. The MCC is a comprehensive performance indicator that takes into account TP, TN, FP, and FN to reduce the randomness and imbalance of classification results (Zhu, 2020).

## 3 Results

### 3.1 Performance of ARE method

To verify the necessity of using the ARE method to enhance the results of paddy rice mapping, we conducted a comparison between the results obtained using ARE and those obtained from overlay or single temporal methods (Fig.4). Our findings indicate that there are significant variations in the results for paddy rice with different phenologies. The single and overlay methods have limitations, while the ARE method can effectively address these limitations by considering the differences in category probability mapping results at different phenological stages. This enhances the accuracy and reliability of paddy rice maps. In addition, the presence of salt and pepper noise in images can significantly impact the accuracy of plot segmentation. However, the ARE method can produce clearer plot edges and demonstrate improved classification performance compared to the results of the paddy rice maps that were overlaid at different periods.

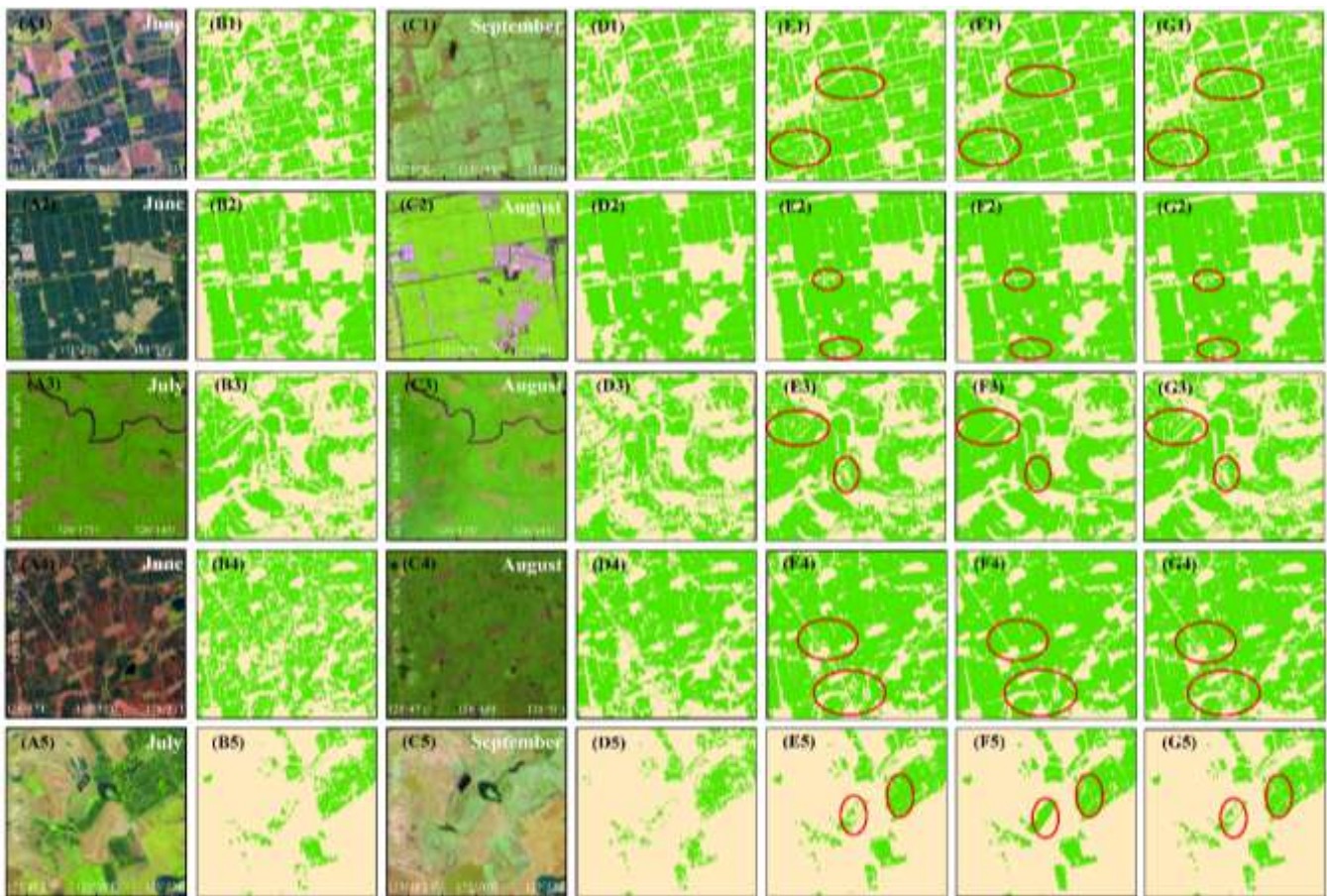

Figure 4: Comparison of paddy rice maps between ARE, single temporal, and overlay methods using Landsat 5 TM and Landsat 8 OLI images. (A1)-(A3), (C1)-(C3) represent pseudo-colored maps of three regions in the Landsat 8 images (bands SWIR 1, NIR, and Red) from June to September. (A4), (A5), (C4), and (C5) represent pseudo-colored maps of two regions in the Landsat 5 images (bands SWIR 1, NIR, and Red) from June to September. $B_i$ display the paddy rice maps corresponding to the images in $A_i$. $D_i$ shows the paddy rice maps corresponding to the images in $C_i$. $E_i$ are the overlay results of the paddy rice maps from $A_i$ and $C_i$. $F_i$ depict ARE paddy rice maps obtained from all images available at that year. $G_i$ depict the ARE paddy rice maps from $A_i$ and $C_i$. $i \in [1,5] \cap \mathbb{Z}$. The red circles in the overlay and ARE maps indicate areas with significant differences in paddy rice.

We chose areas in Northeast China that were captured by multiple Landsat scenes at different growth stages. We also had corresponding ground truth data to quantitatively evaluate the performance of the ARE method. We used a confusion matrix to compare and analyze the performance of the ARE method (Table 2). In comparison to traditional rice mapping methods, the results obtained using the ARE method showed a 6% increase in the OA, 5% increase in the F1 score, and 13% increase in the MCC. These results indicate that the ARE method exhibits higher accuracy and better performance, while the overlay maps and single temporal results demonstrated more severe misclassification and omission.

**Table 2 Confusion matrix for ARE results, overlay results, and single temporal results based on ground truth data.**

| Prediction | Truth | Result 1 Paddy | Result 1 Non-paddy | Result 2 Paddy | Result 2 Non-paddy | Overlay maps Paddy | Overlay maps Non-paddy | ARE maps Paddy | ARE maps Non-paddy |
|---|---|---|---|---|---|---|---|---|---|
| Field survey | Paddy | 0.45 | 0.11 | 0.48 | 0.09 | 0.53 | 0.07 | 0.56 | 0.04 |
| | Non-paddy | 0.16 | 0.28 | 0.14 | 0.29 | 0.08 | 0.32 | 0.06 | 0.34 |
| | UA | 0.80 | 0.64 | 0.84 | 0.67 | 0.88 | 0.80 | 0.93 | 0.85 |
| | PA | 0.74 | 0.72 | 0.77 | 0.76 | 0.87 | 0.82 | 0.90 | 0.89 |
| | OA | 0.73 | | 0.77 | | 0.85 | | 0.90 | |
| | F1 score | 0.77 | | 0.81 | | 0.88 | | 0.92 | |
| | MCC | 0.45 | | 0.53 | | 0.69 | | 0.79 | |
| VHR | Paddy | 0.46 | 0.09 | 0.49 | 0.07 | 0.54 | 0.05 | 0.59 | 0.03 |
| | Non-paddy | 0.19 | 0.26 | 0.15 | 0.29 | 0.11 | 0.30 | 0.06 | 0.32 |
| | UA | 0.84 | 0.58 | 0.88 | 0.66 | 0.92 | 0.73 | 0.95 | 0.84 |
| | PA | 0.71 | 0.74 | 0.77 | 0.81 | 0.83 | 0.86 | 0.91 | 0.91 |
| | OA | 0.72 | | 0.78 | | 0.84 | | 0.91 | |
| | F1 score | 0.77 | | 0.82 | | 0.87 | | 0.93 | |
| | MCC | 0.43 | | 0.55 | | 0.67 | | 0.81 | |
| All | Paddy | 0.46 | 0.10 | 0.49 | 0.08 | 0.54 | 0.06 | 0.58 | 0.03 |
| | Non-paddy | 0.17 | 0.27 | 0.15 | 0.28 | 0.10 | 0.31 | 0.06 | 0.33 |
| | UA | 0.82 | 0.61 | 0.86 | 0.65 | 0.90 | 0.78 | 0.95 | 0.85 |
| | PA | 0.73 | 0.73 | 0.77 | 0.78 | 0.86 | 0.84 | 0.91 | 0.92 |
| | OA | 0.73 | | 0.77 | | 0.85 | | 0.91 | |
| | F1 score | 0.77 | | 0.81 | | 0.88 | | 0.93 | |
| | MCC | 0.45 | | 0.53 | | 0.68 | | 0.81 | |

## 3.2 Performance of the paddy rice results

### 3.2.1 Accuracy evaluation

The performance of the FR-Net and ARE methods for paddy rice mapping was evaluated using confusion matrices and metrics such as UA, PA, F1 score, and MCC based on ground truth data (Table 3). The annual confusion matrix metrics were all found to be at least 0.90 for UA of paddy, PA of paddy and F1 score, and 0.80 for MCC. The overall confusion matrices for the average value of UA of paddy, PA of paddy, OA, F1 score, and MCC are 0.93, 0.91, 0.91, 0.92, and 0.82, respectively. These findings demonstrate the effectiveness of the FR-Net and ARE methods in accurately mapping long-history paddy rice cultivation in Northeast China, confirming their strong generalization across both time and space.

**Table 3 Confusion matrix of paddy rice maps.**

| Year | Validation | Truth Paddy | Truth Non-paddy | UA Paddy | UA Non-paddy | PA Paddy | PA Non-paddy | OA | F1 score | MCC |
|------|-----------|-------|-----------|-------|-----------|-------|-----------|----|----------|-----|
| 2002 | Paddy | 0.49 | 0.03 | 0.94 | 0.90 | 0.91 | 0.93 | 0.92 | 0.92 | 0.84 |
|      | Non-paddy | 0.05 | 0.43 | | | | | | | |
| 2003 | Paddy | 0.61 | 0.03 | 0.95 | 0.86 | 0.92 | 0.91 | 0.92 | 0.94 | 0.83 |
|      | Non-paddy | 0.05 | 0.31 | | | | | | | |
| 2004 | Paddy | 0.49 | 0.05 | 0.91 | 0.89 | 0.91 | 0.89 | 0.90 | 0.91 | 0.80 |
|      | Non-paddy | 0.05 | 0.41 | | | | | | | |
| 2005 | Paddy | 0.46 | 0.05 | 0.90 | 0.92 | 0.92 | 0.90 | 0.91 | 0.91 | 0.82 |
|      | Non-paddy | 0.04 | 0.45 | | | | | | | |
| 2006 | Paddy | 0.48 | 0.05 | 0.91 | 0.91 | 0.92 | 0.90 | 0.91 | 0.91 | 0.82 |
|      | Non-paddy | 0.04 | 0.43 | | | | | | | |
| 2007 | Paddy | 0.52 | 0.04 | 0.99 | 0.89 | 0.91 | 0.91 | 0.91 | 0.92 | 0.82 |
|      | Non-paddy | 0.05 | 0.39 | | | | | | | |
| 2008 | Paddy | 0.54 | 0.04 | 0.93 | 0.88 | 0.92 | 0.90 | 0.91 | 0.92 | 0.81 |
|      | Non-paddy | 0.05 | 0.37 | | | | | | | |
| 2009 | Paddy | 0.47 | 0.04 | 0.90 | 0.90 | 0.90 | 0.92 | 0.91 | 0.92 | 0.82 |
|      | Non-paddy | 0.05 | 0.44 | | | | | | | |
| 2010 | Paddy | 0.47 | 0.05 | 0.90 | 0.90 | 0.90 | 0.90 | 0.90 | 0.90 | 0.80 |
|      | Non-paddy | 0.05 | 0.43 | | | | | | | |
| 2011 | Paddy | 0.46 | 0.04 | 0.92 | 0.90 | 0.90 | 0.92 | 0.91 | 0.91 | 0.81 |
|      | Non-paddy | 0.05 | 0.45 | | | | | | | |
| 2013 | Paddy | 0.52 | 0.04 | 0.93 | 0.89 | 0.91 | 0.91 | 0.91 | 0.92 | 0.82 |
|      | Non-paddy | 0.05 | 0.39 | | | | | | | |
| 2014 | Paddy | 0.58 | 0.04 | 0.94 | 0.87 | 0.92 | 0.89 | 0.91 | 0.93 | 0.81 |
|      | Non-paddy | 0.05 | 0.33 | | | | | | | |
| 2015 | Paddy | 0.60 | 0.03 | 0.95 | 0.86 | 0.92 | 0.91 | 0.92 | 0.94 | 0.83 |
|      | Non-paddy | 0.05 | 0.32 | | | | | | | |
| 2016 | Paddy | 0.54 | 0.04 | 0.93 | 0.88 | 0.92 | 0.90 | 0.91 | 0.92 | 0.81 |
|      | Non-paddy | 0.05 | 0.37 | | | | | | | |
| 2017 | Paddy | 0.62 | 0.03 | 0.95 | 0.83 | 0.91 | 0.91 | 0.91 | 0.93 | 0.80 |
|      | Non- | 0.06 | 0.29 | | | | | | | |

| Year | Type | | | | | | | | | |
|---|---|---|---|---|---|---|---|---|---|---|
| | paddy | | | | | | | | | |
| 2018 | Paddy | 0.61 | 0.03 | 0.95 | 0.86 | 0.92 | 0.91 | 0.92 | 0.94 | 0.83 |
| | Non-paddy | 0.05 | 0.31 | | | | | | | |
| 2019 | Paddy | 0.63 | 0.03 | 0.95 | 0.83 | 0.91 | 0.91 | 0.92 | 0.93 | 0.80 |
| | Non-paddy | 0.06 | 0.29 | | | | | | | |
| 2020 | Paddy | 0.51 | 0.04 | 0.93 | 0.89 | 0.91 | 0.91 | 0.91 | 0.92 | 0.82 |
| | Non-paddy | 0.05 | 0.40 | | | | | | | |
| 2021 | Paddy | 0.56 | 0.04 | 0.93 | 0.90 | 0.93 | 0.90 | 0.92 | 0.93 | 0.83 |
| | Non-paddy | 0.04 | 0.36 | | | | | | | |
| 2022 | Paddy | 0.51 | 0.04 | 0.93 | 0.93 | 0.94 | 0.91 | 0.93 | 0.94 | 0.86 |
| | Non-paddy | 0.03 | 0.42 | | | | | | | |
| 2023 | Paddy | 0.60 | 0.03 | 0.95 | 0.88 | 0.93 | 0.92 | 0.93 | 0.94 | 0.84 |
| | Non-paddy | 0.04 | 0.33 | | | | | | | |
| Overall | Paddy | 0.53 | 0.04 | 0.93 | 0.88 | 0.91 | 0.90 | 0.91 | 0.92 | 0.82 |
| | Non-paddy | 0.05 | 0.38 | | | | | | | |

### 3.2.2 Compare with agricultural statistics

The accuracy of the paddy rice maps we provided has been confirmed using agricultural statistics data. These maps have demonstrated strong consistency and correlation with the agricultural statistics data, as shown in Fig.5. The high $R^2$ value of 0.93 in Fig.5(a) and 0.96 in Fig.5(b) indicates a robust positive correlation between the provided paddy rice maps and the agricultural statistics data. However, the paddy rice cultivation areas were overestimated, as shown in Fig.5(c). This suggests that while these maps provide relatively accurate information on the distribution of paddy cultivation, they cannot be completely accurate. It is essential to consider potential sources of error or uncertainty in both the agricultural statistics data and the mapping methodology, including factors such as data collection methods, spatial resolution, and limitations of the statistical models used. Despite these potential limitations, Figure 5 provides strong evidence supporting the credibility of the provided paddy rice maps.

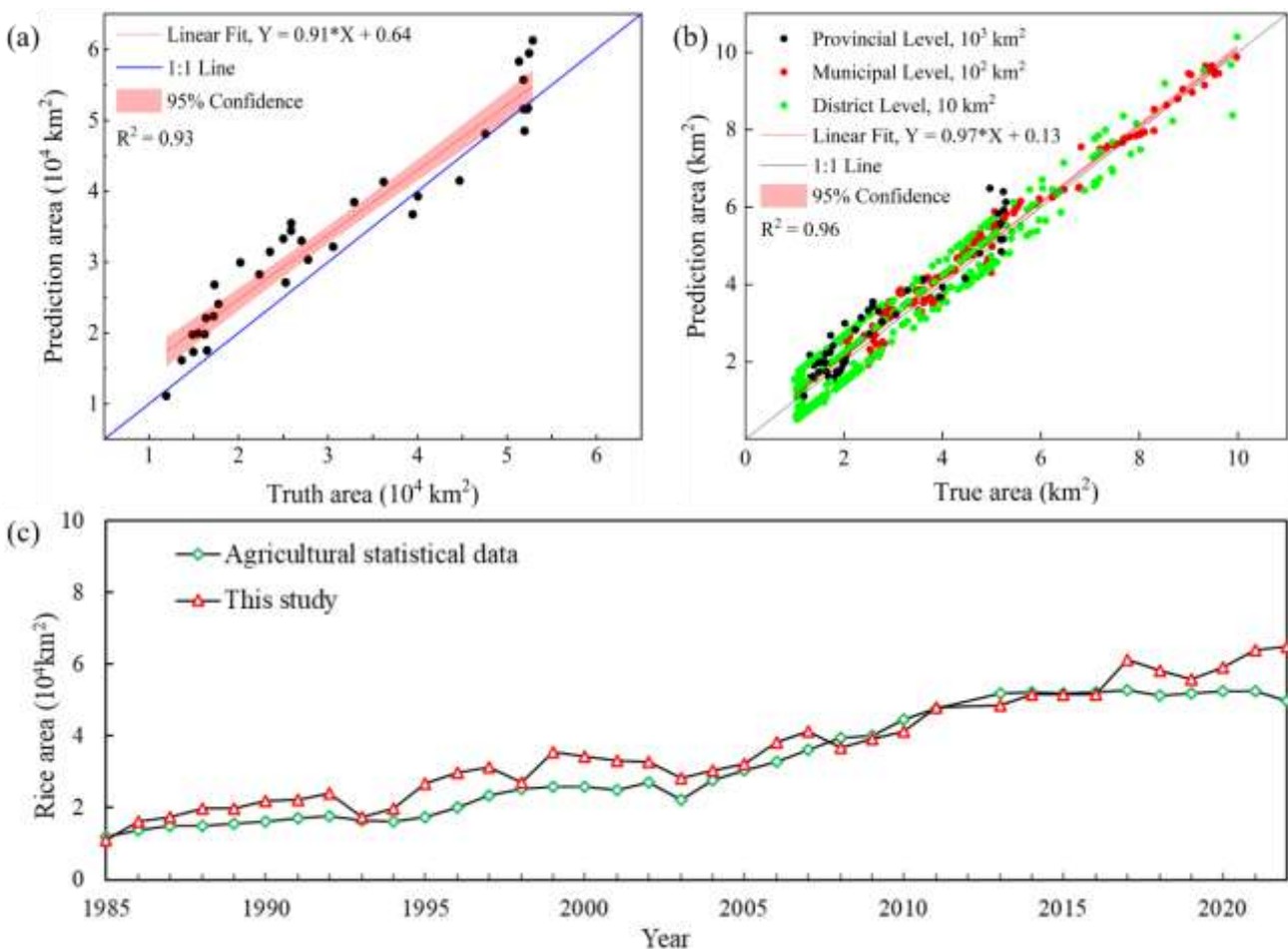

285 **Figure 5: Comparison of paddy rice maps with agricultural statistical data. (a) presents the paddy rice area comparison across the entire study area. (b) is the comparison of the paddy rice area at provincial, municipal, and district levels within the study area. (c) shows the difference between the agricultural statistical data and the results of this study regarding paddy rice area estimation.**

### 3.2.3 Comparison with other paddy rice datasets

In this study, we compared the paddy rice products with other representative products (Fig.6). These included products
290 generated by the coarse resolution of MODIS data (Liu et al., 2018), the phenology-based method using optical Landsat data (Zhang et al., 2023), and the combined use of Sentinel-1/2 SAR and optical data (Shen et al., 2023). The results showed that the paddy rice products obtained from MODIS data overestimation due to fragmented plots and mixed pixels in rice cultivation areas, as shown in Fig.6(C1)-(C4). Similarly, the paddy rice products generated using the phenology-based method with optical Landsat data demonstrated omissions, as illustrated in Fig.6(D1)-(D4). In contrast, the results of this study, shown in Fig.6(B1)-
295 (B4), reveal that most paddy pixels were well recognized using the FR-Net and ARE methods. Furthermore, the mapping results that combined SAR and optical data also exhibited some leakage of the paddy pixels. The speckle noise from Sentinel-

1 contributed to an increase in salt-and-pepper noise in the paddy rice results, which in turn reduced the mapping accuracy, as illustrated in Fig.6(E1)-(E4). The deep network FR Net and ARE achieved a more complete representation of rice plots with clearer boundaries and improved mapping accuracy compared to individual phenological methods.

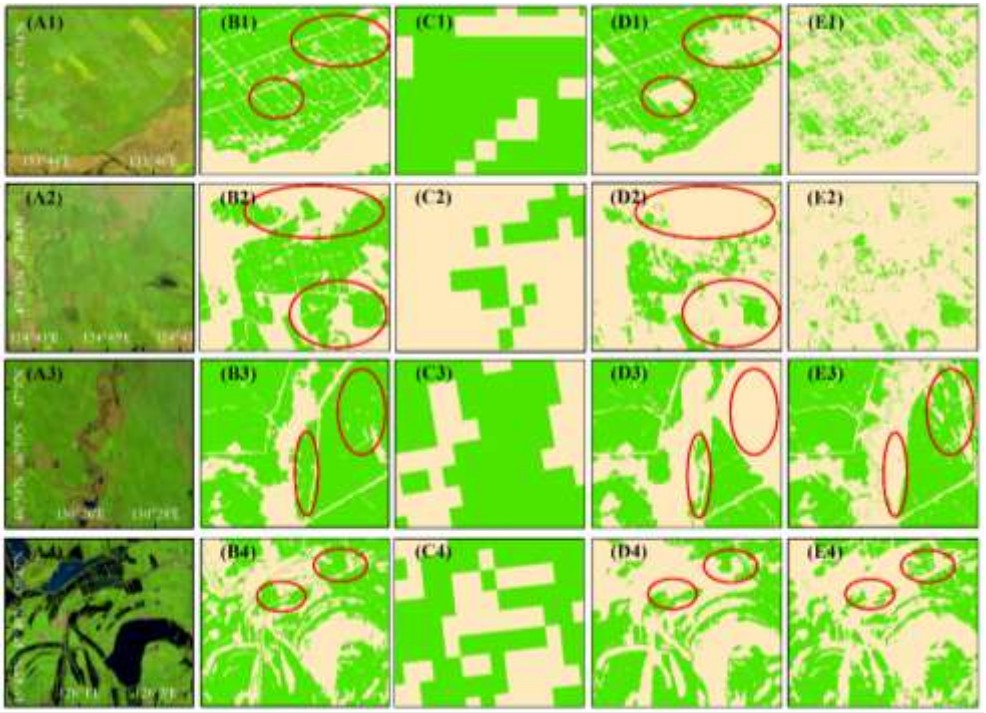

**Figure 6: Comparison of the paddy rice products of this study with the existing products. (A1)-(A4) represents pseudo-colored (bands SWIR 1, NIR, and Red) of Landsat 8/9 images. (B1)-(B4) is the result of this study. (C1)-(C4) shows the paddy rice with MODIS data. (D1)-(D4) demonstrate the paddy rice with Landsat data. (E1)-(E4) depict the paddy rice maps using Sentinel-1 and Sentinel-2 images. The red circles indicate areas with significant differences in paddy rice.**

### 3.3 Spatiotemporal patterns of paddy rice from 1985 to 2023

In the long history of paddy rice mapping, clouds have posed a challenge that hinders the accuracy of the annual mapping results. In certain areas, it is difficult to obtain clear-sky observation data during the growth period of paddy rice, resulting in limited coverage in the annual mapping results. Figure 7 shows the study area, which lacks at least one clean-sky observation during the yearly growth stage of paddy rice. This figure demonstrates that in Northeast China, there are areas where at least one clean-sky observation during the annual growth stage of paddy fields is not possible. To improve the quality of the yearly mapping results in this study, the missing pixels were filled based on good observations from before and after the year, and year-to-year cloud coverage maps were obtained.

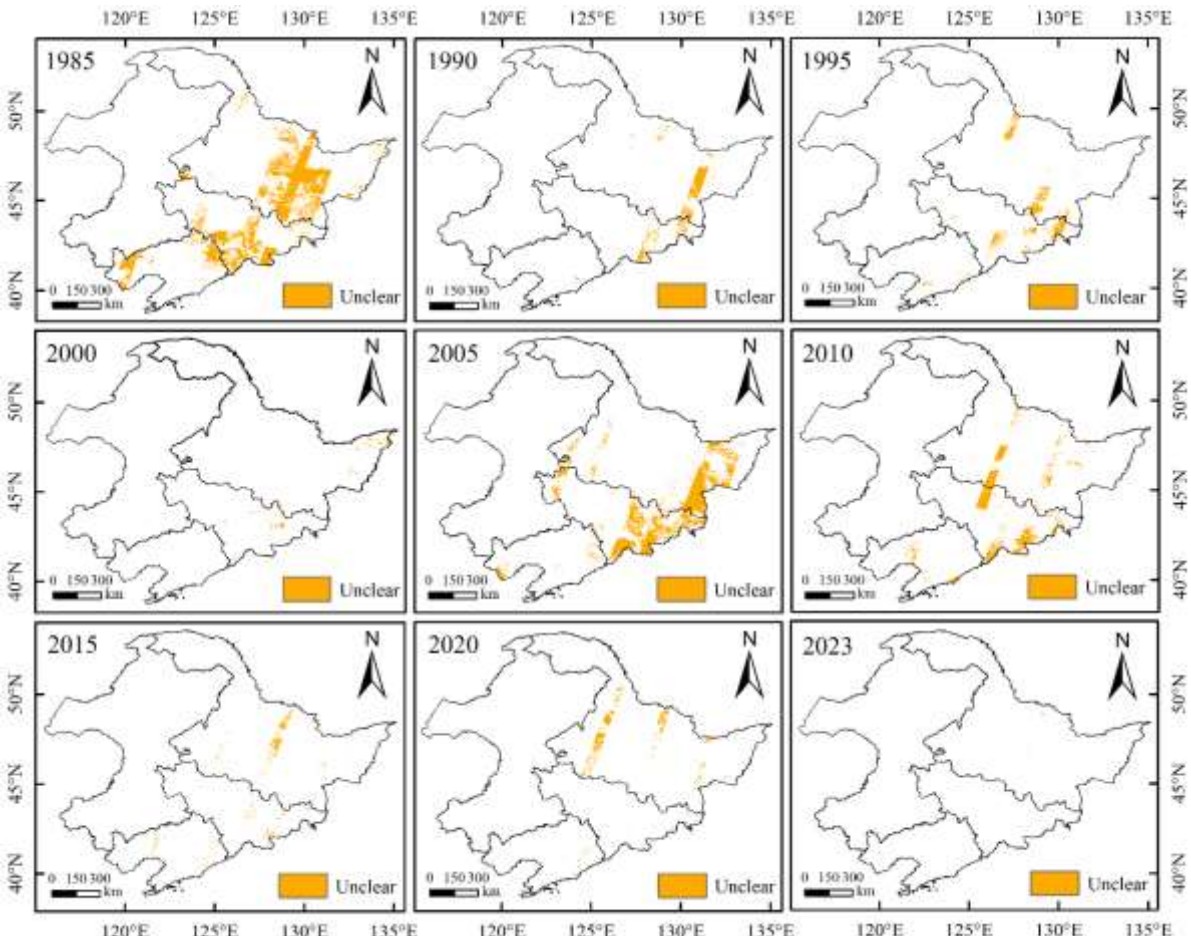

**Figure 7: Areas lack at least one clean-sky observation during the annual growth stage of paddy.**

Figure 8 shows the filled paddy cultivation in Northeast China from 1985 to 2023, mainly distributed along river banks. A noteworthy increase in the area of paddy rice cultivation was observed in the central region of the study area between 1985 and 1990. Additionally, there was a significant expansion of paddy rice cultivation in northeastern Heilongjiang Province from 2005 to 2010. In contrast, the area of rice cultivation in Liaoning Province remained relatively stable throughout the entire period from 1985 to 2023.

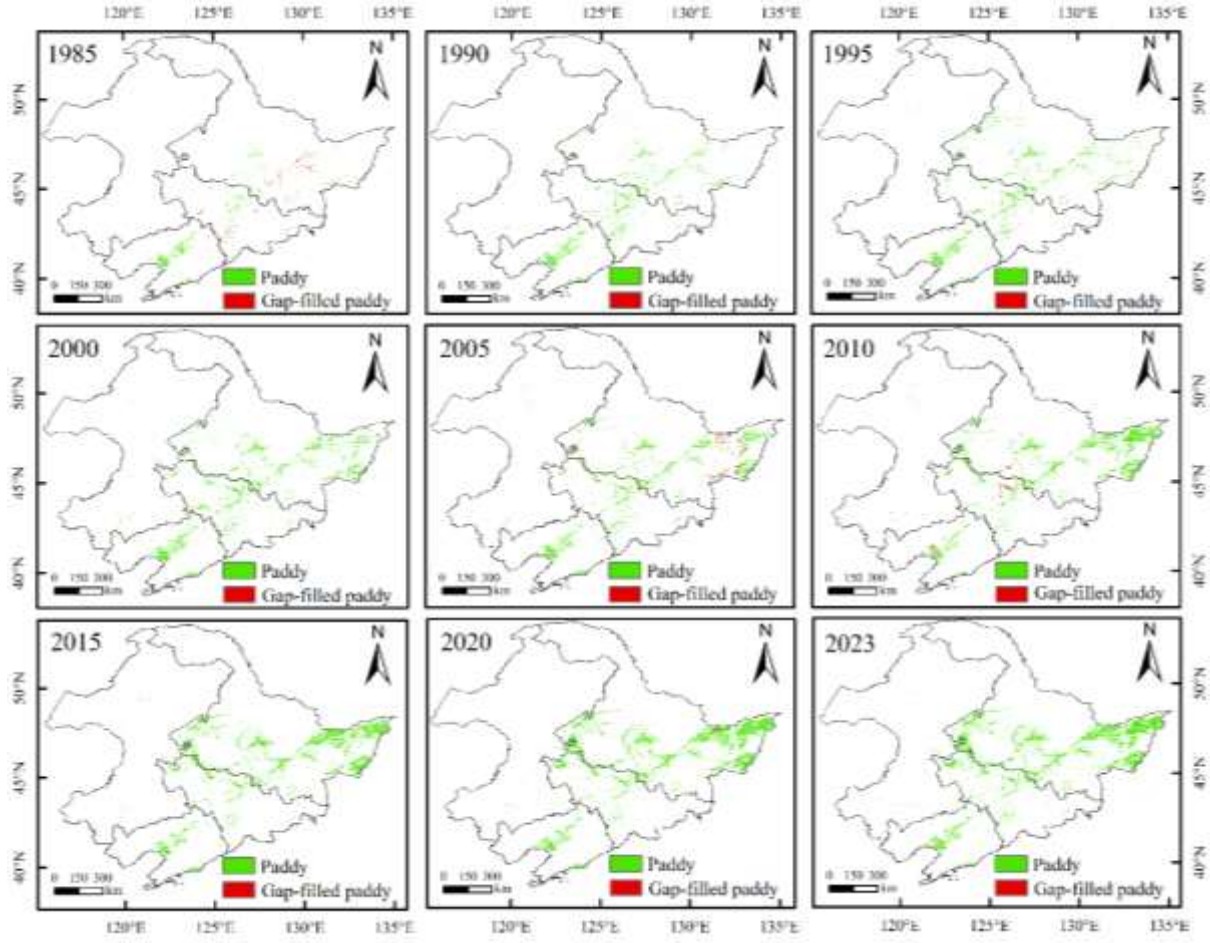


**Figure 8: Spatial distribution of paddy rice in Northeast China.**

We examined the changes in paddy field distribution in Northeast China from 1985 to 2023. The changes are illustrated in Fig.9. The paddy area in the study area exhibited significant growth, with a net increase of $5.34\times10^4$ km². Different provinces exhibited varying patterns of change. Heilongjiang province saw the largest increase in paddy area, with a gain of $4.33\times10^4$ km² from 1985 to 2023, followed by Jilin province with a $0.70\times10^4$ km² increase. Liaoning province and northeastern Inner Mongolia experienced smaller increases, with $0.16\times10^4$ km² and $0.15\times10^4$ km², respectively. This suggests that regions at high latitudes have become more suitable for paddy rice cultivation in recent decades. Additionally, Liaoning province experienced the largest reduction in paddy rice cultivation area, with a decrease of $0.20\times10^4$ km², followed by Jilin province, Heilongjiang province, and northeastern Inner Mongolia with a decrease of $0.09\times10^4$ km², $0.03\times10^4$ km² and $0.01\times10^4$ km², respectively.

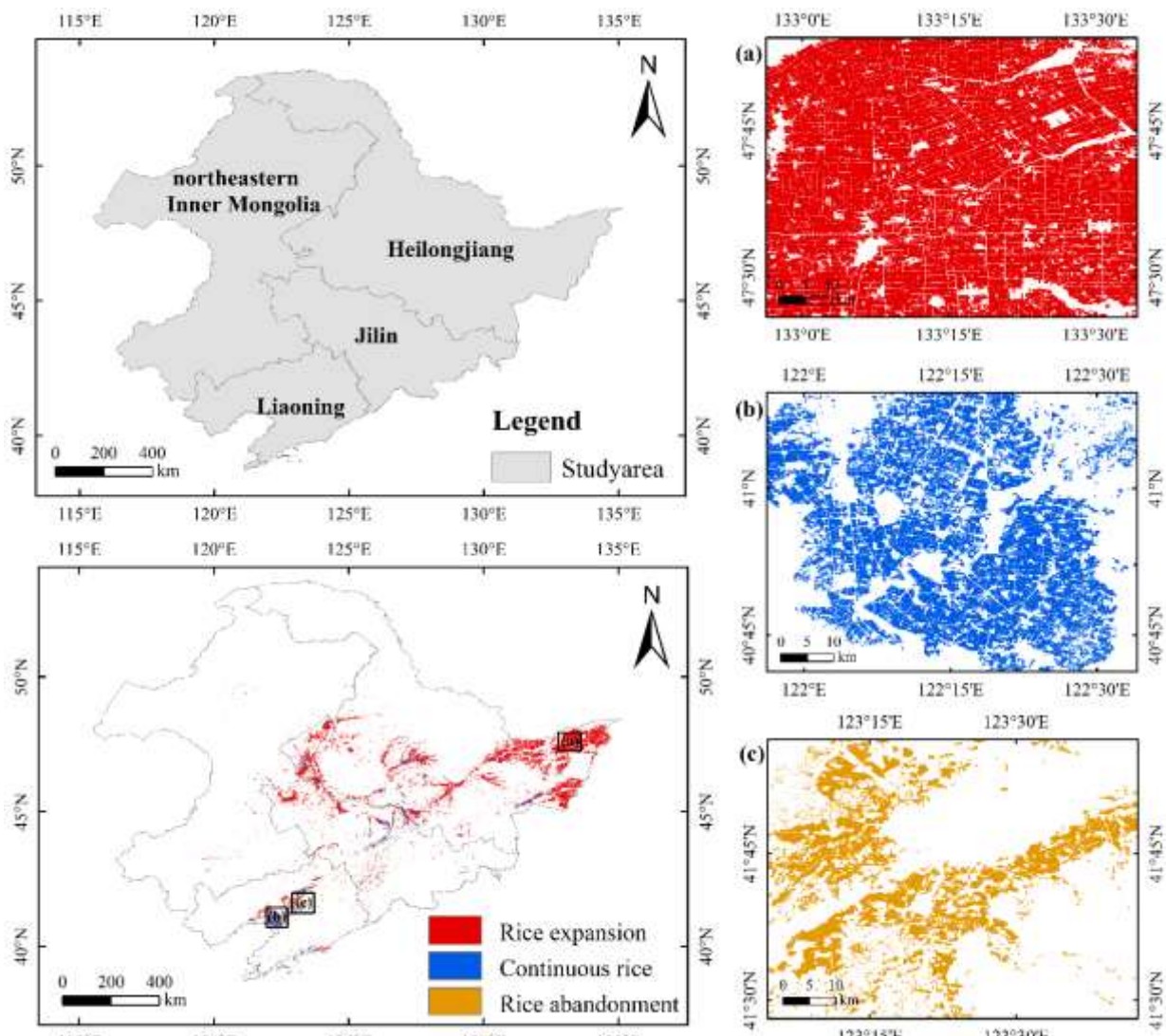

**Figure 9: Trends of paddy rice cultivation area in Northeast China from 1985 to 2023.**

## 4 Discussion

### 4.1 How the cross-sensor training sample impacts the accuracy of long history mapping

The Landsat 8/9 OLI sensor is an improvement and enhancement of the Landsat 5 TM sensor. It shares many characteristics such as the same spatial resolution, scene size, etc. However, there are slight differences in spectral range and sensor radiation calibration quality. These differences lead to variations in feature distribution, impacting the accuracy of crop mapping. This study focuses on the impact of these differences on paddy rice mapping accuracy. We conducted a comprehensive paddy rice

mapping using an across-sensor training dataset to understand how these differences affect the accuracy of the model and how

we can combine this dataset to achieve the best mapping accuracy. In this study, we employed nine different combinations of training and test sets to assess the performance of the cross-sensor dataset for paddy rice mapping in Northeast China (Table 4).

**Table 4 Performance of different combinations of training and test datasets.**

| Combination Num. | Training set | Test set | F1 score | UA | PA |
|---|---|---|---|---|---|
| 1 | Landsat 5 | Landsat 5 | 0.85 | 0.83 | 0.89 |
| 2 | Landsat 5 | Landsat 8 | 0.48 | 0.46 | 0.53 |
| 3 | Landsat 8 | Landsat 5 | 0.53 | 0.51 | 0.57 |
| 4 | Landsat 8 | Landsat 8 | 0.86 | 0.84 | 0.90 |
| 5 | Landsat 5 | Landsat 5+Landsat 8 | 0.62 | 0.61 | 0.65 |
| 6 | Landsat 8 | Landsat 5+Landsat 8 | 0.64 | 0.62 | 0.67 |
| 7 | Landsat 5+Landsat 8 | Landsat 5+Landsat 8 | 0.84 | 0.82 | 0.88 |
| 8 | Landsat 5 (Transfer learning with Landsat 8) | Landsat 8 | 0.67 | 0.65 | 0.70 |
| 9 | Landsat 8 (Transfer learning with Landsat 5) | Landsat 5 | 0.70 | 0.68 | 0.74 |

In Table 4, we observed varying classification accuracy performances when using different combinations of training and test

datasets. Combinations with minimal differences in feature distribution between the training and test datasets tended to have higher mapping accuracy. For example, combination numbers 1 and 4 achieved the highest values for the F1 score, UA, and PA. However, combination numbers 2 and 3, which had significant feature distribution differences between the training and test datasets, exhibited the lowest values of F1 score, UA, and PA. This means models trained using single-sensor images cannot be effectively applied to images from other sensors (combination numbers 2 and 3). Therefore, to ensure reliable large-

scale mapping and longitudinal consistency, we advocate for systematic cross-sensor fusion strategies rather than single-sensor dependencies, thereby mitigating the differences in feature distributions between sensors.

Furthermore, we implemented transfer learning by utilizing partial Landsat 8/9 OLI or Landsat 5 TM data as the training set to fine-tune the models developed using combination numbers 1 or 4. Specifically, 20% of the Landsat 8/9 OLI or Landsat 5 TM images were selected as the training set to fine-tune the model originally trained with a training sample of combination

numbers 1 or 4. For example, the model trained with combination number 1 was further trained using 20% of Landsat 8/9 OLI images to achieve subtle adjustments in the model weights, which were then applied to the remaining Landsat 8/9 OLI images. The results revealed that fine-tuning the models enhanced accuracy compared to the combination of numbers 2 and 3. However, the accuracy of the fine-tuned models was lower than that of the models trained using a cross-sensor dataset, or the same sensor data, i.e., combination numbers 1, 4, and 7.

In summary, due to differences in spectral ranges, spectral characteristics, and spectral responses between the Landsat 5 TM and Landsat 8/9 OLI sensors, it is essential to use sufficient numbers of training data to achieve fine-mapping results. Moreover, while model fine-tuning can improve accuracy and generalization ability, it may not be effective for crop mapping in this case, especially when dealing with a long history and data from different sensors.

**4.2 Contribution of ARE method in overcoming errors caused by feature differences in training and test dataset**

Landsat images exhibit significant differences in spectral and spatial characteristics during the different growth periods of paddy rice. However, it is challenging to gather sufficient training samples to cover the entire growth period of paddy rice on a large scale and over an extended period. Consequently, the trained model may struggle to learn the features present in the test dataset on a large scale, leading to significant differences in the mapping results of paddy rice at different phenological stages. Figure 10 illustrates the difference in paddy rice results between the overlay and ARE methods. Compared to the ARE

result, the overlay method exhibits a phenomenon of misclassification (Fig.10 (b)). As shown in Fig.10 (a) and (c), pixels with a lower category probability mean a significant difference between the learned features of FR-Net and the features of the predicted images. These uncertainties were inherited by the overlay method, but the ARE method selected the best category probability and thus presented fewer false alarms or omissions, as shown in Fig.10 (d).

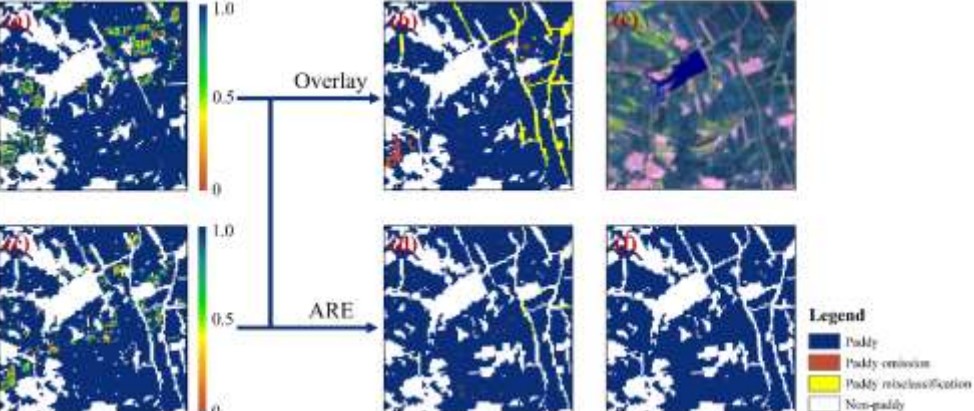

**Figure 10: Comparison of paddy rice results between overlay and ARE results. (a) and (c) show the category possibilities for identifying paddy rice at different stages. (b) presents the overlay result, while (d) displays the ARE result. (e) depicts the pseudo-colored map, and (f) is the true map of paddy rice based on manual interpretation.**

    To assess the capability of ARE in overcoming the differences in features between training and test samples in different phenological periods and regions, we used the Landsat images with a Path/Row of 120/028 in 2016 as an example. We

evaluated the UA, PA, and F1 scores of paddy rice mappings from individual phenological periods and the ARE paddy rice map (Table 5). In this work, we used the image acquired on May 31$^{st}$ as the training set, and the images from other periods were individually used as the test set to demonstrate the effectiveness of the ARE method in enhancing paddy rice mapping results at different phenological stages.

**Table 5 Comparison of accuracy among single phenological period maps and the ARE paddy rice map.**

| Training image date | Testing image date | Matrices | | |
| --- | --- | --- | --- | --- |
| | | UA | PA | F1 score |
| 31/05 | 16 June 2016 | 0.66 | 0.95 | 0.74 |
| | 02 July 2016 | 0.62 | 0.87 | 0.72 |
| | 18 July 2016 | 0.59 | 0.80 | 0.68 |
| | 03 August 2016 | 0.58 | 0.77 | 0.67 |
| | 19 August 2016 | 0.63 | 0.92 | 0.71 |
| | 20 September 2016 | 0.52 | 0.66 | 0.64 |
| ARE map | | 0.92 | 0.94 | 0.93 |

Moreover, we separately utilized Landsat images from different regions in the same year as the test set to evaluate the accuracy of the FR-Net model and then compared the accuracy of paddy rice maps from different regions within the same year with those achieved by the ARE method. The Landsat images with different Path/Row in 2016 were used as the inputs to calculate the UA, PA, and F1 scores of FR-Net (Table 6).

**Table 6 Comparison of accuracy among the paddy rice maps with different Path/Row and the ARE paddy rice map.**

| Training image Path/Row | Testing image Path/Row | Matrices | | |
| --- | --- | --- | --- | --- |
| | | UA | PA | F1 score |
| 116/026 | 116/028 | 0.68 | 0.58 | 0.59 |
| | 120/026 | 0.32 | 0.44 | 0.37 |
| | 120/027 | 0.71 | 0.79 | 0.73 |
| | 120/028 | 0.70 | 0.64 | 0.65 |
| | 120/029 | 0.15 | 0.34 | 0.17 |
| | 122/030 | 0.06 | 0.21 | 0.09 |
| ARE map | | 0.91 | 0.93 | 0.92 |

Tables 5 and 6 confirmed the benefits of using the ARE method to significantly improve the temporal and spatial mapping accuracy of FR-Net for paddy rice. Additionally, optimizing the mappings of paddy rice based on multiple phenological periods can help diminish the impact of cloud cover on satellite image quality. Therefore, introducing the ARE method is crucial for enhancing the accuracy of paddy rice maps in large-scale and long history mapping.

## 4.3 Uncertainties in paddy rice classification

The results and comparison analysis presented in this study demonstrate that the proposed ARE method effectively improves annual mapping accuracy by using category probability at different phenological periods. However, several limitations should be acknowledged. First, despite the confusion matrix evaluation indicators were calculated using the proportion of area, uncertainties persist due to the influence of mixed pixels. Second, the ARE method improves mapping accuracy by processing

multiple classification category probability of FR-Net, but its effectiveness is limited when only a single image is available.

Furthermore, the efficacy of the ARE method is intricately linked to the quantity and quality of the training samples. When the quantity and quality of training samples are limited, the confidence of the category probabilities of the paddy rice mapping results integrated into the ARE method is low, and the improvement in precision that the ARE method can provide may be insignificant. Third, we evaluated the applicability of the ARE method using the 2020 paddy rice mapping results as a case study. Regions where the ARE method was prone to classification errors (omission or commission) under specific conditions were identified, and their spatial distribution is shown in Fig.11. These regions cover an area of approximately 179 km$^2$, accounting for 0.014% of the total study area. For these areas, the final classification was determined based on the maximum class probability derived from images across different phenological stages within the same year. Finally, due to the time restrictions of field investigations and the availability of Google Earth data, this study only validated the paddy rice mapping results from 1985 to 2001 using agricultural statistical data, which may reduce the confidence level in the long history mapping. Moreover, the non-random availability of Google Earth imagery across different years remains an inherent limitation. Nonetheless, we adhered to the principle of random sampling by utilizing all available imagery from the crop growing season, and aimed to validate the mapping results through as random a sampling process as possible. Therefore, continuous improvement of methods and techniques is essential to enhance the reliability of paddy rice mappings in light of the current issues associated with ARE methods.

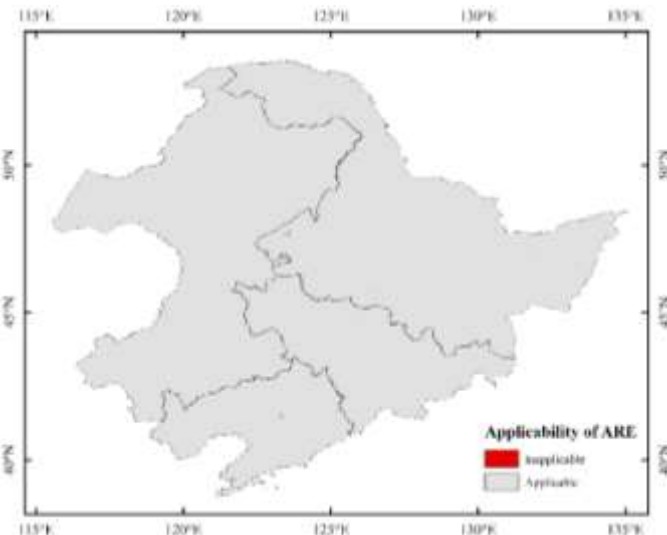

**Figure 11: Applicable region of the ARE method.**

## 5  Code/Data availability

The FR-Net codes are available at https://github.com/xialang2012/Paddy. The paddy rice maps produced with 30 m resolution in this study are accessible at https://doi.org/10.6084/m9.figshare.27604839.v1 (Zhang et al., 2024). The dataset includes a set of GeoTIFF images in the ESPG: 4326 spatial reference system. The values 1 and 0 represent paddy and non-paddy,

respectively. We encourage users to independently verify the paddy rice maps. In addition, Landsat 5 TM and Landsat 8/9 OLI are available on United States Geological Survey (USGS) (https://earthexplorer.usgs.gov).

## 6 Conclusions

This study developed a cross-sensor training dataset for recognizing paddy rice in Northeast China from 1985 to 2023 at a 30-meter spatial resolution. The presented annual result enhancement (ARE) method, which considers the differences in category probability of FR-Net at different growth stages to alleviate the impact of the limited training sample in large-scale and across-sensor paddy rice mapping. The ARE method showed a 5% increase in both the UA and PA, 6% increase in the OA, 5% increase in the F1 score, and 13% increase in the MCC than the traditional rice mapping method. The overall mapping results

obtained from the FR-Net and ARE methods achieved high average values of UA of paddy, PA of paddy, OA, F1 score, and MCC values of 0.93, 0.91, 0.91, 0.92, and 0.82 respectively. The overall trend from 1985 to 2023 indicated a significant increase of $5.34 \times 10^4$ km$^2$ in the rice cultivation area, expanded by 4.81 times. In all, this study confirmed that the semantic segmentation network FR-Net and ARE methods could accurately realize the large-scale and long-history monitoring of paddy rice, and it would provide a paddy rice classification dataset to detect the spatiotemporal patterns and dynamic mechanisms of

paddy rice.

## Author contributions

Z.Z., L.X., and F.Z. designed the study and the methodology, Z.Z. and L.X. wrote the code and generated the data, Y.G., J.Y., and Y.Z. provided the ground truth data, Z.Z., S.W., and P.Y. and checked samples and evaluate the resulting maps. All authors analyzed the data, wrote, and edited the manuscript.

## Competing interests


The authors declare that they have no known competing financial interests or personal relationships that could have appeared to influence the work reported in this paper.

## Acknowledgments

This work was supported by the National Natural Science Foundation of China (42401448/42301442), the National Key

Research and Development Program of China (2022YFD2001105), Fundamental Research Funds for Central Non-profit Scientific Institution (1610132025003), the open project of State Key Laboratory of Efficient Utilization of Arid and Semi-arid Arable Land in Northern China, the Institute of Agricultural Resources and Regional Planning, Chinese Academy of Agricultural Sciences (EUAL-2023-02), and the Agricultural Science and Technology Innovation Project of the Chinese

Academy of Agriculture Sciences. The authors are grateful to the United States Geological Survey (USGS) for providing Landsat data.

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
