# Peer review of "Long history paddy rice mapping across Northeast China with deep learning and annual result enhancement method"

_Earth System Science Data, 2024_

## Author Response (AR1)

**Author's Thanks:**

We sincerely appreciate the reviewer's dedicated time and expertise in critically evaluating our work. The constructive feedback has prompted essential refinements to both the scholarly substance and structural clarity of this manuscript, significantly elevating its academic contribution. Below we provide a systematic point-by-point response to each comment. The italicized content represents the modifications made in the manuscript.

**Response to RC1**

**RC1 Comment 1:**

Field data is critical for model training and map validation. However, the manuscript lacks clarity and details regarding how the authors collected the field samples and how they chose the field target for visual interpretation through GEE. The sampling strategy for field data collection is unknown, especially for validation data collection. Using points to validate 30-m maps is inappropriate especially when mixed pixels occur. The reported accuracies could be largely impacted by 30-m mixed pixels when only point data are employed for map validation.

**Author's Response 1:**

Thank you for your valuable suggestion. To ensure comprehensive representation of the validation dataset, the study area was stratified into major paddy cultivation areas and non-paddy regions. Ground truth data were randomly sampled from each stratum, maintaining an approximate 3:2 ratio between paddy and non-paddy areas. This resulted in a total of 68856 paddy samples and 39098 non-paddy samples, as illustrated in Fig.1(b). Specifically, the dataset includes 21254 paddy samples and 13160 non-paddy samples obtained from field survey, complemented by 47602 paddy samples and 25938 non-paddy samples from Google Earth VHR imagery. To mitigate the effects of mixed pixels in the validation results, detailed field observation data was conducted within a 30m radius around each sample point, recording the proportions of coverage for paddy rice. And the confusion matrix of was calculated in terms of the proportion of area. This comprehensive dataset was used to evaluate the accuracy and reliability of the paddy rice mapping results.

**RC1 Comment 2:**

For the map evaluation, the distribution of the validation dataset is not reported in the manuscript. Are they derived from probability sample? Otherwise, the validation would not be valid.

Constructing a confusion matrix based on pixel counting is not recommended. The population error matrix of classes with cell entries should be expressed in terms of the proportion of area. Besides, the uncertainty of these accuracy metrics should also be reported. Refer to Olofsson et. al (2014) for a guideline on how to conduct map accuracy evaluation solidly.

**Author's Response 2:**

Thanks for your valuable comment. The validation dataset was developed through a stratified random sampling approach. To reduce the influence of mixed pixels (paddy and other types) and according to the methodology proposed by Olofsson et al., (2014) we refined the confusion matrices presented in Tables 2 and 3 to reflect area-based proportions. Furthermore, detailed descriptions of the evaluation metrics and their corresponding formulas have been incorporated into Section 2.5, while the report of accuracy indicator uncertainties has been included in Section 4.3. Besides, Tables 4, 5, and 6 in the Discussion section evaluates the accuracy of FR-Net models based on training set, no ground truth data; therefore, no changes are needed.

Reference:

Olofsson, P., Foody, G.M., Herold, M., Stehman, S.V., Woodcock, C.E., Wulder, M.A., 2014. Good practices for estimating area and assessing accuracy of land change. Remote Sens. Environ. 148, 42–57.

**RC1 Comment 3:**

L22: Are there average numbers from 1985 to 2023? Make it clearer.

**Author's Response 3:**

Thanks for your valuable comment. These values represent the average accuracy evaluation indicators for paddy rice maps from 1985 to 2023, and we have clarified these in the manuscript. The modified content is as follows: "*The overall mapping result obtained from the FR-Net model and ARE methods achieved high average values of user accuracy (UA) of paddy, producer accuracy (PA) of paddy, overall accuracy (OA), F1 score, and Matthews correlation coefficient (MCC) values of 0.93, 0.91, 0.91, 0.92, and 0.82, respectively.*"

**RC1 Comment 4:**

L44-45: What is the justification for such a statement that it's challenging to produce long-term maps using phenology-based methods? any references?

**Author's Response 4:**

Thanks for your valuable comment, we have modified the corresponding content in the manuscript. The modified content is as follows: "*While this method is simple and effective, cloud cover can cause gaps in satellite data during the growth period, increasing the difficulty to track phenological changes accurately. This limitation can reduce the accuracy of long-term rice mapping. (Carrasco et al., 2022; Dong et al., 2015).*"

Reference:

Carrasco, L., Fujita, G., Kito, K., Miyashita, T., 2022. Historical mapping of rice fields in Japan using phenology and temporally aggregated Landsat images in Google Earth Engine. ISPRS J. Photogramm. Remote Sens. 191, 277–289.

Dong, J., Xiao, X., Kou, W., Qin, Y., Zhang, G., Li, L., Jin, C., Zhou, Y., Wang, J., Biradar, C., Liu, J., Moore, B., 2015. Tracking the dynamics of paddy rice planting area in 1986–2010 through time series Landsat images and phenology-based algorithms. Remote Sens. Environ. 160, 99–113.

**RC1 Comment 5:**

L64: I think you are saying that the final map for a specific year is derived from multiple intermediate maps within the year. But 'multiple annual results" means multiple yearly maps, i.e., a map for each year. This could be confusing.

**Author's Response 5:**

Thank you very much. We believe 'intermediate maps' is a better option. According to your guidance, we have revised it in the manuscript. As below: "However, determining the final mapping result from multiple annual results remains a challenge for large-scale paddy rice mapping." to "*However, determining the final mapping result for a specific year from multiple intermediate maps remains a challenge for large-scale paddy rice mapping.*"

**RC1 Comment 6:**

L116: You mentioned Result_pre in the text but there is no Result_pre in Eq.1.

**Author's Response 6:**

Thanks for your valuable comment. We are sorry for the careless, and the Eq. (1) has been revised accordingly.

$$\text{"}Result_{pre} = \begin{cases} nonpaddy. else \\ paddy, \forall i \in [1, \dots m]: Result_i = paddy \end{cases} \text{'} \tag{1}\text{"}$$

**RC1 Comment 7:**

L132: From Eq.2, t represents the image corresponding to the highest absolute value of the

difference between the category probability and 0.5, not the direct highest Pi. Why not use max (Pi) instead? For example, if P1=0.1, P2=0.6, then there would be t=1, and Pt=P1=0.1. Would you determine the final results as non-paddy since Pt < 0.5?

**Author's Response 7:**

Thanks for your valuable comment. If P1=0.1, P2=0.6, we would determine the final results as non-paddy. This is because we set 0.5 as the classification threshold. |P1-0.5|=0.4 indicates a higher confidence level in classifying the sample as non-paddy, whereas |P2-0.5|=0.1 suggests a lower confidence level in identifying it as paddy, accompanied by higher classification uncertainty. To enhance the robustness of the classification result, we prioritize the feature with stronger confidence. Consequently, the final category is determined as non-paddy in this scenario. If max ($P_i$) is used, misclassification issues may occur, such as incorrectly identifying non-paddy as paddy. For example, if P1=0.1, P2=0.2, and P3=0.6, the category probability of being non-paddy is higher. In this case, using max($P_i$) to classify the final result as paddy would be erroneous.

**RC1 Comment 8:**

L136: Are you using exactly the same parameters (models) in these different phenological stages? Otherwise, how could you ensure that the category probability outputs among m images are comparable?

**Author's Response 8:**

Thanks for your valuable comment. We share the same point. Exactly the same parameters (models) were used across all phenological stages. Only in this way can we ensure the category probability outputs of the m images are comparable. We supplemented it in the manuscript: "*In this study, all FR-Net models utilized the same set of parameters for subsequent analysis.*" Thank you.

**RC1 Comment 9:**

L142: In which months did you download the data and from where? Specify the date range for each year, or the same range across years if they are consistent.

**Author's Response 9:**

Thanks for the insightful comment. We obtained Landsat Collection 2 Level-2 surface reflectance products for Northeast China from the United States Geological Survey (USGS), covering the period from May to September annually between 1985 and 2023. A total of 13,809 images were downloaded, each with a spatial resolution of 30 meters. The aforementioned content has been

incorporated into the manuscript.

**RC1 Comment 10:**

L145: Please provide the specific band names instead of numeric names.

**Author's Response 10:**

Thanks for your valuable comment, we have modified the corresponding content in the manuscript.

"*In addition, we selected the Blue, Green, Red, Near Infrared (NIR), Shortwave Infrared (SWIR) 1, and Shortwave Infrared (SWIR) 2 bands of Landsat 8/9 OLI and Blue, Green, Red, NIR, SWIR 1, and SWIR 2 bands of Landsat 5 TM images to map the paddy rice.*"

**RC1 Comment 11:**

L156: There are some issues with the validation dataset. The sampling strategy is unknown. How did you select the field sites to visit? If the distribution of validation datasets is biased (not randomly selected), then the map accuracy based on the validation datasets is not valid. Did you collect field data as point observations? Using points to validate 30-m maps is inappropriate especially when mixed pixels occur. What are the spatial and temporal distributions of training and validation data?

**Author's Response 11:**

Thanks for your valuable comment. To ensure the accuracy of the validation dataset, field samples were collected in major paddy cultivation areas and non-paddy regions. The spatial distribution of these validation samples is depicted in Figure 1(b), with stratified random sampling applied to randomly select samples within each stratum. A total of 34414 field survey data and 73540 VHR data were collected as point observations. The sampling time of field and VHR data aligned with the Landsat image acquisition period (May to September) to ensure temporal consistency. To mitigate mixed pixels effects in validation results, detailed field observations were conducted within a 30m radius around each sample point, recording the proportion coverage of paddy rice. And the confusion matrix of was calculated in terms of the proportion of area.

**RC1 Comment 12:**

L170: 29906 + 9968 + 50956 + 16985 = 107815, this is less than the total size (68865 + 39098 = 107963, L154), did you remove any ground samples and why? What are the criteria for dividing the entire ground data into these training/validation sets with these specific numbers, for Landsat5 and 8/9 respectively?

**Author's Response 12:**

Thanks for your valuable comment. In this study, no ground samples were removed. The validation data (ground truth) comprised 107963 field survey and Very High Resolution (VHR) samples, while 107815 Landsat images, each with a size of 256×256 pixels, were used to train the model.

**RC1 Comment 13:**

L205: Please use explicit band names. In Fig.3, did you use the same probability threshold of 0.5 in both overlay maps and the ARE maps? For the red circled area, e.g., in E5, a non-paddy pixel in the overlay map means all category probability outputs are less than 0.5. According to Eq2, for ARE methods, a paddy pixel must have a probability greater than 0.5. How come a non-paddy pixel in the overlay map would become a paddy pixel in the ARE map?

**Author's Response 13:**

Thanks for your valuable comment, we have modified explicit band names in Lines 205-206. In Fig.3, we used the same probability threshold of 0.5 in both overlay maps and the ARE maps. Regarding E5 in Fig.3, due to manuscript length constraints and to emphasize the differences between the overlay results and ARE results, only the results from columns A and C were displayed. However, the results in column F were generated using all available images from May to September for that year. To further clarify, we have added column G, which depicts the ARE results calculated from the results in columns A and C.

[Figure]

**Figure 3: Comparison of paddy rice maps between ARE, single temporal, and overlay methods using Landsat**

**5 TM and Landsat 8 OLI images. (A1)-(A3), (C1)-(C3) represent pseudo-colored maps of three regions in the Landsat 8 images (bands SWIR 1, NIR, and Red) from June to September. (A4), (A5), (C4), and (C5) represent pseudo-colored maps of two regions in the Landsat 5 images (bands SWIR 1, NIR, and Red) from June to September. $B_i$ display the paddy rice maps corresponding to the images of $A_i$. $D_i$ shows the paddy rice maps corresponding to the images of $C_i$. $E_i$ are the overlay results of the paddy rice maps from $A_i$ and $C_i$. $F_i$ depict ARE paddy rice maps obtained from all images available at that year. $G_i$ depict the ARE paddy rice maps from $A_i$ and $C_i$, $i \in [1,5] \cap \mathbb{Z}$. The red circles in the overlay and ARE maps indicate areas with significant differences in paddy rice.**

**RC1 Comment 14:**

L212: The distribution of the validation points is totally unknown. Are they derived from probability samples? Otherwise, the validation would not be valid. A confusion matrix based on pixel counting is not recommended. The population error matrix of classes with cell entries should be expressed in terms of the proportion of area. Besides, the uncertainty of these accuracy metrics should also be reported. Refer to Olofsson et. al (2014) for a guideline on how to conduct map accuracy evaluation in a solid manner.

Reference:

Olofsson, P., Foody, G.M., Herold, M., Stehman, S.V., Woodcock, C.E. and Wulder, M.A., 2014. Good practices for estimating area and assessing accuracy of land change. Remote sensing of Environment, 148, pp.42-57.

**Author's Response 14:**

Thanks for your valuable comment. We employed a stratified random sampling strategy to select ground validation samples across major paddy cultivation areas and non-paddy areas within the study area, with a total of 107954 samples. Furthermore, to account for the influence of mixed pixels, we reformulated the confusion matrix based on the proportion of area according to the study of Olofsson (Olofsson et al., 2014), which has been meticulously revised in the manuscript.

Reference:

Olofsson, P., Foody, G.M., Herold, M., Stehman, S.V., Woodcock, C.E., Wulder, M.A., 2014. Good practices for estimating area and assessing accuracy of land change. Remote Sens. Environ. 148, 42–57.

**RC1 Comment 15:**

L236: Fig.4, what is the scale of this comparison? I assume this is the total area in the entire study area. What about the comparisons at the district, municipal, and provincial levels since you collected

the agricultural statistics?

**Author's Response 15:**

Thanks for your valuable comment. In Fig.4, we validated the total area of paddy rice from 1985 to 2023 using agricultural statistical data across the entire study area. Owing to the unavailability of statistical data for certain counties and municipalities, we utilized all publicly available agricultural statistical data from the study area to enhance the validation of paddy rice mapping results at the provincial, municipal, and district levels in the manuscript.

[Figure]

**Figure 4: Comparison of paddy rice maps with agricultural statistical data. (a) presents the paddy rice area comparison across the entire study area. (b) is the comparison of the paddy rice area at provincial, municipal, and district levels within the study area. (c) shows the difference between the agricultural statistical data and the results of this study regarding paddy rice area estimation.**

**RC1 Comment 16:**

L263: what if there are no clear-sky observations available in the proceeding and subsequent years? did you leave it as no data? In Fig.7, what is "interpolated paddy"? Did you interpolate your classification directly from the classification in the previous/next year, if there is no cloud-free satellite data in the current year? This has to be clarified.

I assume the interpolated paddy pixels are derived from satellite data that are interpolated from previous/next Landsat observations. The term 'interpolated paddy' implies that the classified pixel

itself is somehow interpolated, which makes no sense.

**Author's Response 16:**

Thank you for your valuable comment, we have modified the corresponding content in the manuscript. After analyzing and processing the Landsat images from 1985 to 2023 in Northeast China, no sustained deficiency in clear-sky observation data availability has been identified within the study area over the examined period. If the method proposed in this study is applied to other regions facing this issue, incorporating multi-source data such as MODIS, Sentinel-1, and Sentinel-2 could be considered to achieve a more comprehensive analysis. And the term 'interpolated paddy' refers to the paddy rice result derived using a multi-year comprehensive method, based on the historical phenological patterns from the nearest available clear-sky year's image. In addition, to avoid ambiguity, we replaced 'interpolated paddy' with 'gap-filled paddy' throughout the manuscript.

**RC1 Comment 17:**

L275-276: There is no figure showing the admin boundaries and labels. It's hard to reader unfamiliar with China to link your descriptive context to the spatial locations in the map.

**Author's Response 17:**

Thanks for your valuable comment, we have modified Figure 8 in the manuscript.

[Figure]

**Figure 8: Trends of paddy rice cultivation area in Northeast China from 1985 to 2023.**

**RC1 Comment 18:**

L301: This is contradictory to the table. Instead, #1 and #4 show that using data from only one sensor achieved the best accuracy and had the best results (if the models are trained and applied to the same sensor), compared to other scenarios using multiple sensors.

**Author's Response 18:**

Thanks for your valuable comment, we have revised the corresponding content in the manuscript. The modified content follows: "*This means models trained using single-sensor images cannot be effectively applied to images from other sensors (combination numbers 2 and 3). Therefore, to ensure reliable large-scale mapping and longitudinal consistency, we advocate for systematic cross-sensor fusion strategies rather than single-sensor dependencies, thereby mitigating the differences in feature distributions between sensors.*"

**RC1 Comment 19:**

L304-305: Not clear how you conducted 'transfer learning". For #8, training on L5 & 20% L8, then apply to L8? Need to clarify.

**Author's Response 19:**

Thanks for your valuable comment. Sorry for the misleading, and we have clarified the issue you raised in the manuscript.

For #8, the model trained with combination number 1 was further trained using 20% of Landsat 8/9 OLI images to achieve subtle adjustments in the model weights, which were then applied to the remaining Landsat 8/9 OLI images.

**RC1 Comment 20:**

L306: An enhanced accuracy compared to what?

**Author's Response 20:**

Thanks for your valuable comment, we have modified the corresponding content in the manuscript. "The results revealed that fine-tuning the models enhanced accuracy compared to combination numbers 2 and 3."

**Response to RC2**

**RC2 Comment 1:**

Abstract: Lines 13–14 provide a good overview of the methodology, including the data and model used. However, the workflow of ARE is unclear to readers unfamiliar with it. Adding one or two sentences explaining its mechanism would be beneficial. The explanation in lines 17–18 is too general. Readers may struggle to grasp the concept just from the abstract. Line 22 presents rice expansion values but lacks a specified duration. I recommend revising this sentence and the following one for better readability and clarity.

**Author's Response 1:**

Thank you for your valuable comment. To improve readability, we have supplemented an explanation of the ARE method. The corresponding content is as follows: "*ARE integrates differences in category probability and confidence levels of the FR-Net across phenological stages, effectively reducing classification uncertainty. This approach could mitigate the impact of limited training sample on large-scale and across-sensors paddy rice mapping.*"

In addition, we have specified the time range of expansion in paddy rice cultivation area in the manuscript, as follows: "*The study revealed that the area used for paddy rice cultivation in Northeast China increased from $1.11 \times 10^4$ km² to $6.45 \times 10^4$ km² between 1985 and 2023.*"

**RC2 Comment 2:**

Introduction: This section is well-structured and easy to follow. In line 63, the phrase "multiple annual results" is unclear. Using plain language would improve readability.

**Author's Response 2:**

According to your suggestion, we have revised "However, determining the final mapping result from multiple annual results remains a challenge for large-scale paddy rice mapping." to *"However, determining the final mapping result for a specific year from multiple intermediate maps remains a challenge for large-scale paddy rice mapping."* in the manuscript. Thanks for your valuable comment.

**RC2 Comment 3:**

Methods: This section requires substantial improvement for better clarity. Adding a workflow diagram would significantly enhance readability and coherence. The sections are loosely connected.

The authors should clearly specify: Which datasets were used to feed the model? Which results were used for ARE? Which datasets were used for validation and how they relate to the modeling outputs or ARE results?

**Author's Response 3:**

Thank you for your significant comment. We have integrated a workflow diagram into the manuscript and provided a concise description of the process. Additionally, we have revised the text to clarify the role of each dataset in the respective sections, with enhanced explanations of their relationships with the modeling outputs and ARE results. The supplementary details are summarized below:

*2.2.1 Workflow of the study*

*The workflow for paddy rice mapping in Northeast China is illustrated in Figure 2. Firstly, Landsat data, paddy and non-paddy samples derived from Google Earth data and field survey, and the paddy cultivation area from agricultural statistics data between 1985 and 2023 were compiled to validate the accuracy of the annual paddy rice mapping results in this study. Secondly, a cross-sensor dataset containing 115 scenes paddy rice maps was generated using the XGBoost classifier combined with manual visual correction. This cross-sensor dataset served as training and testing datasets for developing the FR-Net model. Thirdly, based on the cross-sensor dataset, we employed the FR-Net and ARE methods to account for category probability differences across different phenological periods within a year to reconstruct annual paddy rice maps for Northeast China from 1985 to 2023, and the paddy rice maps were systematically validated with validation data. Finally, the paddy rice mapping results in this study were compared with representative products, and we analyzed the spatial and temporal dynamic characteristics of paddy rice in Northeast China.*

[Figure]

*Figure 2: Workflow of the study.*

**Which datasets were used to feed the model and how they relate to the modeling outputs or ARE results?**

**Response:** The 115 scenes cross-sensor paddy rice maps, generated by the XGBoost classifier and refined through manual visual correction, were fed into the FR-Net model as input data.

**Which results were used for ARE and how they relate to the modeling outputs or ARE results?**

**Response:** The ARE framework enhances all intermediate maps generated by the FR-Net model within a year.

**Which datasets were used for validation and how they relate to the modeling outputs or ARE results?**

**Response:** The paddy and non-paddy samples derived from Google Earth and field surveys were utilized to validate the accuracy of ARE results, while agricultural statistical data were employed to assess the accuracy of the ARE results at the district, municipal, provincial, and entire study area levels. Besides, we have supplemented verification of paddy rice cultivation areas at the district, municipal, and provincial levels in the manuscript.

**RC2 Comment 4:**

2.2.1: As this is the most critical section for mapping, more details on FR-Net are necessary. Although Xia et al. (2022) describes the model, a concise explanation of its working principles, strengths, and weaknesses is still needed. This will provide readers with a foundational understanding, allowing them to refer to the cited work for further details.

**Author's Response 4:**

Thank you for your important comment. We have added a concise explanation of FR-Net, including working principles, strengths, and weaknesses. The revised content is as follows: *The multi-resolution feature fusion unit (MRFU) serves as the core component of FR-Net, specifically designed to achieve high-resolution semantic segmentation while maintaining precise output quality. The MRFU regulates feature propagation through controlled information flow, integrates multi-scale feature representations via resolution-specific streams, and preserves spatial fidelity through hierarchical resolution retention. Its architecture comprises two distinct pathways: the horizontal stream, which preserves native resolution through identity mapping operations, and the vertical stream, which doubles channel capacity while halving spatial resolution. Additionally, the framework incorporates 3×3 convolutional layers with a stride of 2, a batch normalization (BN) layer, and a rectified linear unit (ReLU) activation layer. These components work together to control and fuse feature streams with different resolutions. FR-Net has a simple structure and requires minimal computational resources, making it suitable for extracting characteristic information from Landsat data and mitigating the issue of gradient disappearance. However, despite its straightforward design, the cascading operation of multi-resolution feature fusion may result in computational delays, which could hinder its ability to meet the near real-time requirements for agricultural monitoring.*

**RC2 Comment 5:**

2.2.2: In line 113, it would be helpful to first explain why multiple mapping results exist and how

they are produced. The ARE method is not clearly explained, which raises concerns. Equation 2 suggests that a good map (Pt > 0.5) wins only when it has a greater distance from 0.5. However, this approach may not be fair in all situations. For instance, probability values at the start and end of the growing season may be less reliable than those in mid-season, potentially leading to misclassification of rice pixels as non-rice. More details on this method and additional case studies under different conditions would be beneficial.

**Author's Response 5:**

Thank you for your valuable comment. We have explained the reasons for the existence of multiple mapping results and how they were generated, the content is as follows: "*During the growth period of paddy rice, multiple Landsat images of the same area can be obtained, allowing for the identification of each image and resulting in multiple mapping results.*".

In addition, we have supplemented the explanation of ARE method, the content is as follows: "*Therefore, based on the distinct differences in spectral and texture characteristics of paddy rice across growth stages, we developed an annual result enhancement (ARE) method to address this limitation. ARE integrates differences in category probability and confidence levels of the FR-Net across phenological stages, effectively reducing classification uncertainty. This approach mitigates the impact of limited training sample on large-scale and across-sensors paddy rice mapping.*"

We agree with you on the probability values at the start and end of the growing season may be less reliable than those in mid-season. To minimize the impact of the start and end of the rice growing season on mapping accuracy, we selected Landsat images from May to September for paddy rice mapping.

**RC2 Comment 6:**

2.3.1: How is the growing season defined? How is the model trained by using these bands? The band meaning and numbers vary across years and satellite products, how are they handled? I suggest merging 2.3.3 to this section.

**Author's Response 6:**

Thank you for your valuable suggestion. We agreed to consolidated Section 2.3.3 into Section 2.3.1 to enhance structural coherence, as recommended in the review feedback, and we modified the title of 2.3.1 to "Acquisition and processing of Landsat images".

**How is the growing season defined?**

**Response:** The growing season for paddy rice was defined based on region-specific phenological patterns and time-series spectral signatures derived from Landsat images. Specifically, this season spans from pre-transplanting flooding to post-harvest senescence, and in the study its May to September.

**How is the model trained by using these bands?**

**Response:** We selected the Blue, Green, Red, Near Infrared (NIR), Shortwave Infrared (SWIR) 1, and Shortwave Infrared (SWIR) 2 bands of Landsat 8/9 OLI and Landsat 5 TM images to train the FR-Net model.

**The band meaning and numbers vary across years and satellite products, how are they handled?**

**Response:** We rotated and patched these bands data as inputs for XGBoost and FR-Net models, and used them for subsequent analysis.

**RC2 Comment 7:**

2.3.2: This dataset is crucial for validating the study's product and holds significant value for broader research communities in the study area. It is necessary to publish the relevant dataset for validation checking and a broader use.

**Author's Response 7:**

Thank you for your important comment. This study leverages a multi-temporal ground truth dataset to validate the accuracy of our paddy rice mapping product. We are aware that the validation dataset is invaluable for assessing the study's product and holds significant importance for broader research communities in Northeast China. The validation dataset used in this study was contributed by multiple collaborating institutions (including the authors' affiliations), and we have not obtained explicit authorization from other participating entities to publicly release the complete dataset. Researchers who require access for academic purposes may contact the corresponding author to request a subset of the validation data.

**RC2 Comment 8:**

2.3.3: This section is very confusing and needs to be recontructed. First, what is the connect of XGBoost to the DL model? Given it can generate the paddy and non-paddy maps, what are the differences between its results and the DL model? Second, The ROIs in Fig 1(c)&(d) are very large. From my understand, they indicate paddy and non-paddy. Does it mean that within the ROI, all

pixels are either paddy or non-paddy? Third, how was the manual correction conducted? Forth, What does the mask mean in line 167?

**Author's Response 8:**

Thank you for your valuable comment.

**First, what is the connect of XGBoost to the DL model? Given it can generate the paddy and non-paddy maps, what are the differences between its results and the DL model?**

**Response:** The XGBoost classifier generates preliminary rice classification results based on visually interpreted Regions of Interest (ROIs). This initial output undergoes manual rectification to address commission errors (e.g., non-paddy rice areas misclassified as paddy rice) and omission errors (e.g., undetected paddy rice cultivation areas). The DL (FR-Net) model utilizes expert-refined paddy rice results derived from manual correction of XGBoost-generated preliminary maps. Specifically, the XGBoost results after manual corrected serve as training inputs for the DL model, while the DL outputs are further processed through ARE method to generate the final paddy rice product presented in this study.

**Second, The ROIs in Fig 1(c)&(d) are very large. From my understand, they indicate paddy and non-paddy. Does it mean that within the ROI, all pixels are either paddy or non-paddy?**

**Response:** Fig.1(c) and Fig.1(d) present the paddy rice mapping results for 115 scenes across different geographical regions and times based on XGBoost classification with subsequent manual visual correction. The regions displayed in these figures correspond to the spatial positions of input data within the DL model, which contain both paddy rice pixels and non-paddy rice pixels.

**Third, how was the manual correction conducted?**

**Response:** manual correction is the process of manually modifying the misclassification and omission errors of paddy rice to make them the correct category.

**Forth, What does the mask mean in line 167?**

**Response:** the term 'mask' in the manuscript refers to the processed paddy rice classification results derived from 115 original Landsat images through XGBoost classifier and manual visual correction. The raw satellite imagery and corresponding paddy rice masks underwent standardized preprocessing operations including geometric rotation and patching, ultimately generating paired 256×256 pixel patches. These aligned image-mask pairs maintain spatial correspondence between the preprocessed Landsat images (images) and their associated paddy rice mapping results (masks).

Furthermore, we have merged 2.3.3 to 2.3.1, and modified the title of 2.3.1 to "Acquisition and processing of Landsat images".

**RC2 Comment 9:**

2.5: There are no clear criteria for model constraints, such as loss functions. This should be explicitly mentioned.

**Author's Response 9:**

Thank you for your valuable comment. We have added the loss function in section 2.5 of this manuscript. The content is as follows: "*To mitigate class imbalance during model training, the Dice loss function was employed in this study. This metric, derived from the Dice similarity coefficient (DSC; Eq.3), demonstrates inherent robustness against skewed class distributions by equivalently weighting false positive and false negative errors during optimization, thereby addressing prevalent challenges in imbalanced semantic segmentation tasks.*

$$DSC = \frac{2 \times |A \cap B|}{|A| + |B|}, \hspace{4cm} (3)$$

*where $|A \cap B|$ quantifies the intersection cardinality between the predicted paddy rice pixels (A) and ground truth paddy rice pixels (B); |A| and |B| represent the quantity of paddy rice pixels in A and B, respectively.*"

**RC2 Comment 10:**

Technique comments:

The authors need to update the caption for figures. What is the scale of the dots in Fig 4, district, county, or province? In (a), the dots are for one year or multiple years? In Fig 7, what is the difference between paddy and interpolated paddy? In Fig 8, what does the trend mean? Are the values on map the change rate?

**Author's Response 10:**

Thank you for your valuable comment.

**What is the scale of the dots in Fig 4, district, county, or province?**

**Response:** In Fig.4, we validated the total area of paddy rice from 1985 to 2023 using agricultural statistical data across the entire study area. In addition, to further confirm the accuracy of the paddy rice maps presented in this study, we utilized all publicly available agricultural statistical data from the study area to validate the paddy rice mapping results at the provincial, municipal, and district

levels in the manuscript.

**In (a), the dots are for one year or multiple years?**

**Response:** The dots in Fig.4(a) represent data for individual years.

**In Fig 7, what is the difference between paddy and interpolated paddy?**

**Response:** In Fig.7, the term 'paddy' denotes the paddy rice result directly classified using clear-sky observations, while the term 'interpolated paddy' refers to the paddy rice result derived using a multi-year comprehensive method, based on the historical phenological patterns from the nearest available clear-sky year's image. To avoid ambiguity, we have replaced 'interpolated paddy' with 'gap-filled paddy' throughout the manuscript.

**In Fig 8, what does the trend mean? Are the values on map the change rate?**

Response: In Fig.8, the trend means the whether there have been changes in the paddy rice cultivation areas in 1985 and 2023, and the values on maps refer to the areas where paddy rice cultivation increased, decreased, and remained unchanged in 2023 compared 1985.

---

## Author Response (AR2)

**Author's Thanks:**

We sincerely appreciate the reviewer's dedicated time and expertise in critically evaluating our work. The constructive feedback has prompted essential refinements to both the scholarly substance and structural clarity of this manuscript, significantly elevating its academic contribution. Below we provide a systematic point-by-point response to each comment. The italicized content represents the modifications made in the manuscript.

**Response to Referee #1**

**Comment 1:**

Eq.2 is the most critical part of this study. If P1=0.1, P2=0.2, P3=0.6, P4=0.7, P5=0.8, would you still classify the pixel as non-paddy because |P1-0.5|=0.4? Assuming that P1 and P2 are in the early seasons (e.g., in May) and P3, P4 and P5 are in the peak or late seasons (e.g., August or September), the classification probability for a paddy pixel is expected to increase as crop progresses, because the satellite signals to differentiate paddy/non-paddy become more distinctive in the late seasons compared to that in early seasons. I think it is more reasonable to determine the pixel as paddy in such scenarios. And how do you decide the category if multiple 't' values are found, for example, P1=0.1 and P2=0.9, then |P1-0.5| = |P2-0.5| = 0.4?

**Response 1:**

Thank you for this insightful and critical comment. You have raised an excellent point regarding the limitation of the decision rule in Eq.2. We have thoroughly analyzed the possibility you pointed out and have addressed it in the Discussion section of this manuscript.

We acknowledge that while the scenarios you described may occur theoretically, their practical impact on the final mapping accuracy is minimal. Specifically, as the paddy rice cultivated in Northeast China is transplanted rice, it exhibits a distinct identification feature during the early growth stages—namely, the flooding signal. Therefore, situations where the probability of paddy rice presence is low in the early stage (e.g., P1 = 0.1) but high in the peak growth stage (e.g., P5 = 0.8) are extremely rare in practice.

To address such exceptional cases, we applied screening conditions including $|P_i - 0.5| > |P_j - 0.5|$, and $|P_i - P_j| > 0.5$, *where* $i, j \in [1, ..., m]$ , as well as $|P_i - 0.5| = |P_j - 0.5|$, *where* $i, j \in [1, ..., m]$, to identify regions where the ARE method may perform suboptimally. The

condition $|P_i - P_j| > 0.5$ was used because such cases may lead to misclassification or omission errors when Landsat images from different phenological stages are used to classify the same pixel using the ARE method. Taking the 2020 mapping results of the study area as an example (Fig. 1), these regions comprise only 198910 pixels, covering an area of approximately 179 km², which represents merely 0.014% of the total study area ($1.26 \times 10^6$ km²). For these limited areas, the final mapping results for the year were determined by selecting the maximum class probability from the classification results of images acquired across different phenological stages within the same year.

[Figure]

Figure 1 Applicable region of the ARE method

**Comment 2:**

Line 209: A 30-m x 30-m square is more appropriate than a 30-m radius circle to match the pixels in Landsat. If multiple land cover types co-exist in a mixed sample, how did you determine the final land cover type for validation? Did you use the 50% proportion threshold? This needs to be clarified.

**Response 2:**

Thank you for your comment. We intended to refer to a 30m × 30m square rather than a 30m radius circle. We acknowledge this error in our original description and have corrected it in the manuscript. Regarding the issue of mixed samples, the final land cover type for validation was determined based on the predominant class within the specified area. Given the large field sizes characteristic of Northeast China, a 50% proportion threshold was deemed highly reasonable for identifying the dominant land cover type. This rationale has been clearly explained in the revised manuscript.

**Response to Referee #3**

**Comment 1:**

A key concern raised by other reviewers is about the reliability of the validation procedure. After the authors' clarification, two issues remain. First, the samples interpreted via Google Earth (GE) images might not be entirely accurate, especially since many GE images were acquired in non-growing season. Second, although the random stratified sampling was used to collect the validation samples, the randomness might not be guarantee for each year due to the non-random availability of GE images in each year, which might comprise the comparability of inter-annual accuracy metrics. However, considering the difficulty in obtaining gold-standard validation samples in historical years, I understand these limitations are unavoidable. Nonetheless, I suggest the authors to include some discussions on the remaining issues in the validation procedure.

**Response 1:**

Thanks for your comment. The samples collected via Google Earth imagery were obtained from images during the growing season (May to September), which helps to mitigate interference from non-growing season conditions. Furthermore, regarding the concern over the non-random availability of Google Earth imagery across years, we have added a discussion of this issue in Section 4.3 of the manuscript and marked the corresponding revisions.

**Comment 2:**

Inconsistent metric notation: The accuracy metric indicators are inconsistently labeled (e.g., F1 in Table 5 vs. F1 score in Table 6). Please standardize these labels (including UA, PA, OA, MCC, F1) throughout the manuscript.

**Response 2:**

Thanks for your comment. All relevant indicators throughout the manuscript have been revised for consistency, and the modifications have been marked in this manuscript.

**Comment 3:**

Line 18: The term 'across-sensors' should be either 'across-sensor' or 'across sensors'.

**Response 3:**

We thank the reviewer for this comment. The term on line 18 has been corrected to 'across-sensor'. Additionally, we have performed a full-text search to find and correct all similar instances to ensure

consistency. All modifications have been marked in this manuscript.

**Comment 4:**

Line 398: 'Probability' should be 'probabilities'.

**Response 4:**

Thank you for the correction. We have changed 'Probability' to 'probabilities' on line 398 (now line 400).

---

## Author Response (AR3)

**Author's Thanks:**

We sincerely appreciate the reviewer's dedicated time and expertise in critically evaluating our work. The constructive feedback has prompted essential refinements to both the scholarly substance and structural clarity of this manuscript, significantly elevating its academic contribution. Below we provide a systematic point-by-point response to each comment. The italicized content represents the modifications made in the manuscript.

**Response to Referee #1**

**Comment:**

I only have a minor comment. Please make sure all the figures are presented in high resolution. Most of the figures are not very clear in this revision.

**Response:**

Thank you for your comment. We have noted your feedback and have thoroughly revised all figures in the manuscript to ensure they are presented in high resolution.

**Response to Referee #3**

**Comment:**

The figure quality is poor in the PDF for my reviewing. It can be improved by submiting original figure files from the authors.

**Response:**

Thank you for your comment. We have noted your feedback and have thoroughly revised all figures in the manuscript to ensure they are presented in high resolution.

**Response to Dr. Shvedko**

**Comment:**

Please ensure that the colour schemes used in your maps and charts allow readers with colour vision deficiencies to correctly interpret your findings. Please check your figures using the Coblis – Color Blindness Simulator (https://www.color-blindness.com/coblis-color-blindness-simulator/) and revise the colour schemes accordingly with the next file upload request. -> Fig. 5(b)

**Response:**

Thank you for your important comment. We have revised the color scheme in Fig. 5(b) accordingly.